# Federated Instruction Tuning of LLMs with Domain Coverage Augmentation

## Abstract

Federated Domain-specific Instruction Tuning (FedDIT) utilizes limited cross-client private data together with various strategies of instruction augmentation, ultimately boosting model performance within specific domains. To date, the factors affecting FedDIT remain unclear, and existing instruction augmentation methods primarily focus on the centralized setting without considering distributed environments. Our experiments reveal that the cross-client domain coverage, rather than data heterogeneity, drives model performance in FedDIT. In response, we propose FedDCA, which optimizes domain coverage through greedy client center selection and retrieval-based augmentation. For client-side computational efficiency and system scalability, FedDCA$^*$, the variant of FedDCA, utilizes heterogeneous encoders with server-side feature alignment. Extensive experiments across four distinct domains (code, medical, financial, and mathematical) substantiate the effectiveness of both methods. Additionally, we investigate privacy preservation against memory extraction attacks utilizing various amounts of public data. Results show that there is no significant correlation between the volume of public data and the privacy-preserving capability. However, as the fine-tuning rounds increase, the risk of privacy leakage reduces or converges.

## 1 Introduction

Table 1: Performance (%) of different augmentation settings in each domain after FedDIT, under the default setting elaborated in Appendix A.2. Zero-shot directly inferences without fine-tuning, while the Base Data utilizes only the client's local data for FedDIT. Additionally, we compare FedDCA with other two augmentation strategies performed on the server: random sampling (Random for short) and direct retrieval (Direct Re. for short and is described in Appendix A.4), respectively.

| Domain | Task/Metric | Zero-shot | Base Data | Random | Direct Re. | **FedDCA (ours)** |
|--------|-------------|-----------|-----------|--------|------------|-------------------|
| Code | HumanEval/Pass@1 | 29.88 | **39.03** | 32.93 | 34.14 | 36.58 |
| Med. | MMLU-Med/Acc. | 70.60 | 68.40 | 71.30 | 72.20 | **74.50** |
| Fin. | FPB/Acc. | 55.94 | 58.25 | 64.19 | 66.31 | **67.24** |
|  | FiQA/Acc. | 18.54 | 14.18 | 13.09 | 19.11 | **35.27** |
|  | TFNS/Acc. | 59.21 | 66.62 | 65.53 | 67.62 | **73.32** |
| Math. | GSM8K/Exact Match | 23.27 | 47.46 | 47.38 | 50.87 | **52.46** |

Recently, federated instruction tuning (FedIT) has gained attention as a novel approach that leverages the principles of federated learning (FL) to facilitate collaborative training of large language models (LLM) in distributed environments while maintaining the confidentiality of private data (McMahan et al., 2017; Ye et al., 2024b; Zhang et al., 2023c). This methodology allows for the exchange of model parameters among distributed data holders, thereby achieving a careful balance between privacy preservation and efficient model optimization. Despite the establishment of various FedIT frameworks (Ye et al., 2024b; Kuang et al., 2023; Zhang et al., 2023c), existing literature has not adequately addressed the practical challenges that Federated Domain-specific Instruction Tuning (FedDIT) may encounter in real-world applications. For instance, FedIT generally necessitates

a sufficient amount of instruction data for fine-tuning, which is often a shortage in domain-specific fine-tuning contexts (Zhang et al., 2024b).

In this study, we investigate FedDIT, a novel approach within the FL paradigm aimed at boosting the performance of LLMs in specific domains. Unlike general FedIT, which seeks to enhance model effectiveness across diverse tasks without accounting for local data shortage, FedDIT encounters the unique challenge of clients possessing only a limited quantity of local domain-specific data. To overcome this, FedDIT enrichs the local data through a specific instruction augmentation strategy. This strategic enrichment is crucial for achieving effective instruction tuning and needs to be meticulously designed to avoid performance degradation. Except for the code domain, which primarily adheres to a standardized paradigm, our results reveal that when clients rely solely on their local data for FedDIT, even the presence of high-quality in-domain local data can be insufficient due to limited scale, leading to a decline in performance, as reflected in the underlined values in Table 1. In summary, the goal of FedDIT is to develop a domain-specific LLM that employs collaborative training and instruction augmentation while safeguarding client privacy, thereby ensuring the model's proficiency in executing tasks pertinent to its designated domain.

For simplify and better instruction quality (Zhang et al. (2024b); Toshniwal et al. (2024)), we focus on a specific scenario of FedDIT, where a server-hosted public dataset exists as an abstraction of open-source instruction datasets on the web that encompasses multiple domains. This dataset is utilized for different instruction augmentation strategies, thereby enhancing the model's performance in specific domains. We elaborate the setting of FedDIT in Appendix A.2.

Additionally, the factors affecting FedDIT are still unclear. Compounding this uncertainty, introducing augmented instructions may further complicate results, making it difficult to ascertain effective improvement strategies. Shepherd (Zhang et al., 2023c) approaches this problem from the perspective of heterogeneity, constructing heterogeneity based on the topic of general instruction datasets. It demonstrates that, unlike the consensus of traditional FL, for general FedIT, heterogeneity has a positive effect. By aggregating diverse instructions from clients, the diversity increases, thereby enhancing the model's adaptability to various tasks. However, it just scratches the surface and does not explore issues in FedDIT.

Going one step further, we conduct experiments to unveil a significant finding (Appendix A.3): there is no monotonic correlation between the degree of non-independent and identically distributed (non-iid) and LLM's performance in the context of FedDIT. Inspired by Explore-Instruct (Wan et al., 2023), which shows the potential of domain coverage in domain-specific instruction tuning. The cross-client domain coverage metric is initially defined, followed by an investigation into its impact on FedDIT. Results demonstrate that domain coverage significantly influences model performance in the corresponding domain.

To maximize the cross-client domain coverage without compromising client data privacy, we propose a novel FedDIT algorithm, **Fed**erated Instruction Tuning of LLMs with **D**omain **C**overage **A**ugmentation, termed **FedDCA**. This algorithm employs a greedy client center selection process and implements instruction augmentation through dense retrieval on the server side. The fundamental idea of FedDCA is to select client centers to expand the diversity and coverage of augmented instruction datasets within a specific domain. By strategically optimizing domain coverage at each step, FedDCA efficiently constructs the augmented train set that enhances both the learning and generalization capabilities of the model, leading to superior performance on domain-specific tasks. Furthermore, to mitigate computational overhead on the client side and enhance the system scalability, we propose FedDCA*, which employs heterogeneous encoders of different sizes and capacities. To achieve feature alignment, we train a projector on the server side using public data and employ contrastive learning techniques.

We demonstrate the effectiveness of FedDCA through comprehensive experiments conducted across four domains: code, medical, financial, and mathematical. These are compared against a range of baselines, which can be categorized into unaugmented and augmented methods. In the unaugmented setting, our method is compared with `FedIT`, which includes four orthodox FL techniques: FedAvg (McMahan et al., 2017), FedProx (Li et al., 2020), SCAFFOLD (Karimireddy et al., 2020), and FedAvgM (Hsu et al., 2019). In the augmented setting, we compare FedDCA against methods such as random sampling, direct retrieval, LESS (Xia et al., 2024), and Self-Instruction (Wang et al., 2022). Additionally, we present the performance outcomes of FedDCA when applied under various

FL strategies. We also compare the computational efficiency on the client side between FedDCA and its variant, FedDCA$^*$. For privacy analysis, experiments against memory extraction attacks are conducted to evaluate how different quantities of retrieved public data affect the privacy of client local data. Results indicate that while reliance on local data increases memorization of sensitive information, the risk of privacy leakage diminishes or converges in the augmented setting as the training rounds progress.

The main contribution is as follows:

- We reveal a critical finding: in the context of FedDIT, data heterogeneity has a non-monotonic relationship with model performance. Instead, cross-client domain coverage substantially impacts LLM's effectiveness, as elaborated in Appendix A.3.

- We propose a novel FedDIT algorithm (Section 4), termed FedDCA, aimed at maximizing cross-client domain coverage through greedy client center selection followed by retrieval-based instruction augmentation executed on the server. Additionally, we introduce FedDCA$^*$ to further lessen the client-side computational overhead while enhancing the system scalability. This variant utilizes a heterogeneous encoder structure, paired with a projector on the server side for feature alignment.

- Through extensive experiments (Section 5), we demonstrate the effectiveness of FedDCA and FedDCA$^*$. We also investigate privacy preservation against memory extraction attacks, conducting experiments based on various amounts of public data. Results suggest that the capacity for privacy preservation does not correlate significantly with the quantity of public data. In contrast, the risk of privacy leakage tends to decrease or converge as the fine-tuning rounds increase.

## 2 PRELIMINARIES

FedDIT aims to leverage cross-client private domain-specific instruction data and utilize the multi-domain public data on the server to achieve instruction augmentation, collaboratively enhancing the model's performance in specific domains. Consider $N$ distributed clients, each with local private data $D_k^l$. The server maintains a public dataset $D^p$ that encompasses multiple domains and is responsible for implementing data augmentation strategies and aggregating model parameters received from clients. The training process follows the standard FL protocol (McMahan et al., 2017). For computational efficiency, we adopt Low-Rank Adaption (LoRA) (Hu et al., 2022) as the fine-tuning method, which involves tuning additional parameters $\Delta\phi$ while keeping the pre-trained LLM's parameters $\phi$ frozen. In the initial training phase, the server dispatches $\phi$ and $\Delta\phi$ to each client for $\mathcal{R}$ training rounds. In the $t$-th round, the server sends the aggregated $\Delta\phi^t$ to clients. Clients use $\Delta\phi^t$ to update their local LoRA parameters $\Delta\phi_k^t$ and conduct instruction tuning based on augmented instruction datasets $D_k$. Subsequently, the clients return $\Delta\phi_k^{t+1}$ to the server. The server then aggregates $\{\Delta\phi_k^{t+1} \mid k = 1, \ldots, N\}$ to obtain $\Delta\phi^{t+1}$ for the next round.

## 3 PROBLEM FORMULATION

For better understanding, we list the frequently used notation in Appendix A.12. The objective of FedDIT is to enhance the domain-specific performance of LLMs through FL without sharing private data (Ye et al., 2024b; Zhang et al., 2024b; 2023c). Under the FL framework, suppose we have $N$ clients, where each client $c_k$ has a local dataset $D_k^l$ with its size $N_k^l$, and an augmented dataset $D_k^g$ from the server public dataset $D^p$, respectively. Due to constraints such as memory, computational overhead, and maximum tolerated training time, client $c_k$ can accept at most $N_k^p$ public instructions. Denote the augmentation strategy as $\Lambda$, through which the server performs instruction augmentation on the public dataset. If $\Lambda$ is `null`, it indicates that clients conduct FedDIT solely based on their local private data, which may lead to performance degradation. Conversely, $\Lambda$ may be classified into two categories: (1) focusing exclusively on the client's own local data distribution or (2) considering the cross-client data distribution. In conclusion, the global objective of FedDIT is defined as follows:

$$\underset{\Delta\phi}{\arg\min} \left\{ F(\phi, \Delta\phi) \triangleq \sum_{k=1}^{N} p_k \left( (1 - \alpha_k) F_k \left( \phi, \Delta\phi_k; D_k^l \right) + \alpha_k F_k \left( \phi, \Delta\phi_k; D_k^g \right) \right) \right\}, \quad (1)$$

where the first term and second term denote the accumulated fine-tuning loss computed on the client $c_k$'s local instructions $D_k^l$ and the augmented public instructions $D_k^g$ from the server, respectively. $p_k$ is the weight of the $k$-th client, which is determined by the ratio of $k$-th client's data size to all clients' total data size. $\alpha_k$ is the ratio of the public data amount of its augmented instruction dataset $D_k$, which is computed as $\frac{N_k^p}{N_k^l + N_k^p}$. Eq.1 minimizes the summed empirical loss across clients' augmented instructions to pursue the in-domain utility of the obtained global model. Denote $P$ as the model parameters and $\mathcal{D}$ as a specific dataset, then the client $c_k$'s empirical loss $F_k(\phi, \Delta\phi_k; \mathcal{D})$ base on $\mathcal{D}$ is calculated as:

$$F_k(\phi, \Delta\phi_k; \mathcal{D}) \triangleq \frac{1}{|\mathcal{D}|} \sum_{j=1}^{|\mathcal{D}|} l(P_{\phi + \Delta\phi_k}; x_j), \tag{2}$$

where $x_j \in D, \forall j \in \{1, 2, \ldots, |D|\}$. The instruction tuning loss $l(\cdot; \cdot)$ on a sample $(x, y)$ is defined as $-\sum_{t=1}^{|y|} \log(w(y_t|x, y_{<t}))$, where $x$ is the formated instruction with Alpaca instruction template[1] and $y$ is the corressponding response.

In contrast to the problem formulations in many prior works on FedIT, the primary distinction in the formulation of FedDIT lies in acknowledging the lack of in-domain instructions across clients. Fed-DIT establishes a public multi-domain dataset on the server for domain-specific instruction augmentation by a specific sampling strategy $\Lambda$. Therefore, each client only needs to hold a few high-quality domain-specific private data to collaborate with other clients to obtain a strong domain-specific LLM.

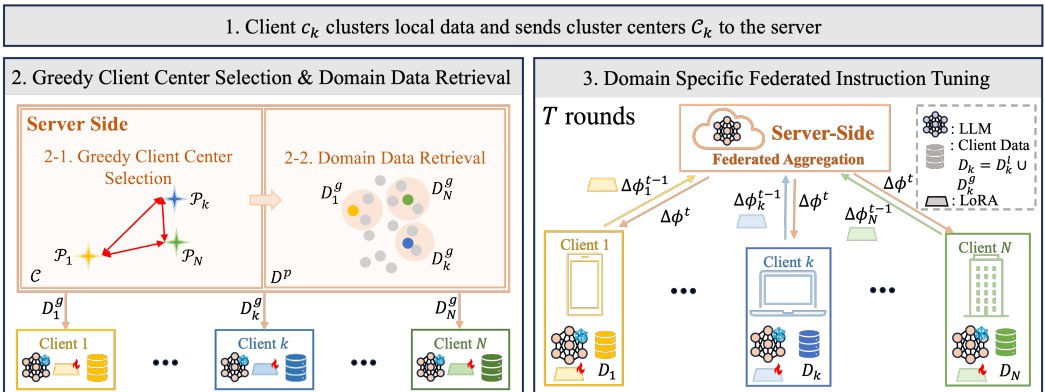

Figure 1: Overview of FedDCA, which consists of three stages: 1) The client $c_k$ performs local instructions clustering and sends cluster centers $\mathcal{C}_k$ to the server. 2) The server first does greedy client center selection to maximize domain coverage and performs client-center-based domain data retrieval, then sends the augmented instructions $D_k^g$ to the client $c_k$. 3) Clients fine-tune the LLM collaboratively, and the LoRA parameters $\Delta\phi$ are exchanged between clients and the server.

## 4 METHOD

We propose Federated Instruction Tuning of LLMs with Domain Coverage Augmentation (Fed-DCA), which enhances domain coverage to obtain a LLM that performs well on domain-specific tasks. FedDCA follows the standard FedIT protocol(Ye et al., 2024b) to perform federated instruction tuning. The novel design of FedDCA lies in the phase before training, which consists of two key modules: greedy client center selection and domain data retrieval, which are elaborated in the following; also see details in Algorithm 1 and Figure 1. For computational efficiency and system scalability, we introduce a variant of FedDCA with heterogeneous encoder setting, named FedDCA*. We first formulate the optimization problem to solve for FedDCA.

---

[1] https://crfm.stanford.edu/2023/03/13/alpaca.html

### 4.1 OPTIMIZATION PROBLEM

As domain coverage directly affects the in-domain performance of the LLM (Appendix A.3), Fed-DCA aims to maximize the domain coverage of the cross-client augmented data $\cup_{i=1}^{N} D_i$ respect to the in-domain data distribution $D^d$, which is defined in Eq. 5. However, as clients can not send the local data to the server for privacy, directly finding a cross-client dataset that maximizes the domain coverage in Eq. 5 is unrealistic. To find a proper client center set $\mathcal{P}$ that maximizes domain coverage and uses it to perform instruction retrieval on the public data is an approximation problem.

Suppose there exists a metric space $\mathcal{X}$, a set of cross-client local instruction embeddings $\mathcal{E} \in \mathcal{X}$, and a set of cluster centers $\mathcal{P} \in \mathcal{X}$. Each client $c_k$ has maximum $\xi$ clusters obtained by k-means algorithm (Wu, 2012). In the federated setting, communication cost is always a critical factor. Consequently, we formulate the optimization problem to include communication costs as follows:

$$\arg\min_{\mathcal{P}} \left\{ \sum_{i=1}^{N} |\mathcal{P}_i| + \sum_{d \in D^d} \left( \min_{p \in \mathcal{P}} sim(d, p) \right) \right\}, \tag{3}$$

s.t. $|\mathcal{P}_i| \leq \xi, \forall i \in \{1, 2, \ldots, N\}$. The second term is the domain coverage of the selected client center set $\mathcal{P}$, and $sim(\cdot, \cdot)$ is the cosine similarity function. However, this optimization problem is NP-hard. Specifically, to select $N$ client center for retrieval, the time complexity is $\mathcal{O}\left( C_{\xi N}^{N} (\mathcal{N}N) \right)$, where $\mathcal{N}$ is the size of the public data $D^p$. As it is a factorial equation, the computational cost explodes with $N$. What's worse, usually, there are enormous amounts of public data on the server, which makes a huge $\mathcal{N}$. For computational efficiency, we propose a greedy algorithm to solve this problem in $\mathcal{O}\left( N\left( N\xi + \xi \log \xi \right) \right)$, which is described below.

### 4.2 GREEDY CLIENT CENTER SELECTION & DOMAIN DATA RETRIEVAL

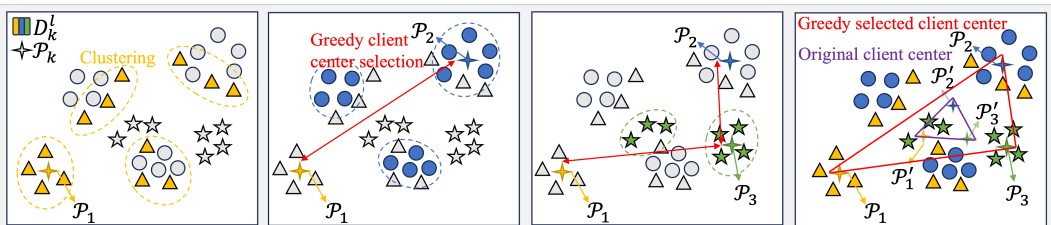

Figure 2: Greedy client center selection iteratively selects the client center in each step, which synthetically considers both the representativeness of the cluster center and its distance from the previously selected client center set $\mathcal{P}$ to maximize the cross-client domain coverage.

As described in Section 4.1, the goal of FedDCA is to find a proper client center set $\mathcal{P}$ that maximizes the domain coverage $d(D^d, \mathcal{P})$, as defined in Eq. 3. For computational efficiency, we propose a greedy algorithm as shown in Algorithm 1 and Figure 2 to solve this problem in polynomial time and obtain a sub-optimal solution. Given the cluster centers $\{\mathcal{C}_i, i = 1, 2, \ldots, N\}$ received from each client, which are obtained by clustering local instructions. FedDCA on the server consists of two main steps: greedy client center selection and client-center-based domain data retrieval.

**Greedy client center selection.** We will consider this problem from two aspects: 1) Select a client center that can represent the distribution of the local data. 2) To optimize the cross-client domain coverage, we filter client centers that are close to the previously selected client centers. First, we randomly choose a client and select the largest cluster center as its center and be the initial of the client center set $\mathcal{P}$. Then, we iteratively select the client center $\mathcal{P}_k$ of the top cluster size while avoiding selecting the cluster center close to the previously selected client center to maximize the domain coverage of $\mathcal{P}$. Specifically, for the $i$-th iteration, we filter $i$-top cluster centers that have the largest summed similarity with previously selected client centers and select the center of the rest largest cluster as the $i$-th client center. This procedure is repeated until we have selected $N$ client centers.

**Domain data retrieval.** For each client center $\mathcal{P}_k$, the server performs dense retrieval (detailed in Appendix A.4) on public dataset $D^p$ to get the top-$N_k^p$ similar public instructions, then sends

---

**Algorithm 1** FedDCA: greedy client center selection & domain data retrieval

---

**Parameters:** Number of clusters $\xi$; Encoder model $w_{enc}$; Client local datasets $D = \{D_1, D_2, \ldots, D_N\}$; Public dataset $D^p$; Similarity score threshold $\alpha$; Number of public data samples retrieved per client $N_k^p$; Client centers $\mathcal{P} = \{p_1, p_2, \ldots, p_N\}$; Pre-encoded public instruction embeddings $\mathcal{E}$.

1: **for** $i \in \{1, 2, \ldots, N\}$ **do**
2:     $\mathcal{E}' \leftarrow \{w_{enc}(x) \mid x \in D_{i,instruction}^l\}$
3:     $\mathcal{C}_i, S_i \leftarrow$ k-means$(\xi)$.fit$(\mathcal{E}')$      ▷ Cluster local instructions, return cluster centers $\mathcal{C}_i$ and sizes $S_i$
4: **end for**
5: Send $\{\mathcal{C}_i, S_i \mid i \in \{1, 2, \ldots, N\}\}$ to the server for the greedy client center selection
6: $\mathcal{P} \leftarrow \{\mathcal{C}_{0,k} \mid k = \arg\max(S_0)\}$      ▷ Initialize the client center set
7: **for** $i \in \{1, 2, \ldots, N-1\}$ **do**      ▷ Greedy client center selection
8:     $\mathcal{S} \leftarrow \sum(\mathcal{C}_i \cdot \mathcal{P}^T, \dim = -1)$      ▷ Compute summed similarity score of each cluster center
9:     $\mathcal{I} \leftarrow (N-i)$- $\arg\text{sort}(\mathcal{S})$      ▷ Filter $i$ cluster centers close to the client center set $\mathcal{P}$
10:     $\mathcal{I}' \leftarrow \arg\text{sort}(-S_i)$      ▷ Sort cluster center by cluster size in descending order
11:     $j \leftarrow$ first element of $\mathcal{I} \cap \mathcal{I}'$      ▷ Selected cluster center $\mathcal{C}_{i,j}$
12:     $\mathcal{P} \leftarrow \mathcal{P} \cup \mathcal{C}_{i,j}$      ▷ Update the client center set
13: **end for**
14: **for** $i \in \{1, 2, \ldots, N\}$ **do**      ▷ Client center based domain data retrieval
15:     $\mathcal{S} \leftarrow \mathcal{P}_i \cdot \mathcal{E}^T$      ▷ Compute similarity score between $\mathcal{P}_i$ and $\mathcal{E}$
16:     $\mathcal{S}' \leftarrow \{s \mid s \in \mathcal{S}, s < \alpha\}$      ▷ Filter instructions with similarity score larger than $\alpha$
17:     $\mathcal{I} \leftarrow$ indices of $N_k^p$-top in $\mathcal{S}'$
18:     $D_i^g \leftarrow \{D_j^p \mid j \in \mathcal{I}\}$
19:     $D_i \leftarrow D_i^l \cup D_i^g$      ▷ Obtain the augmented instruction dataset $D_i$
20: **end for**

---

retrieved public datasets $\{D_1^g, \ldots, D_N^g\}$ to each clients. Specifically, to avoid the overlap between public data and local private data, we set a threshold $\alpha$ to filter the public instructions that have a similarity score larger than $\alpha$ with the client center.

In summary, FedDCA establishes a comprehensive training set that captures the essence and distribution of the domain by iteratively selecting client centers with unique coverage increments. This ensures that selected centers are representative and achieve broad domain coverage. Additionally, domain data retrieval exposes the model to a wider range of in-domain features, enhancing its understanding of domain-specific tasks. This not only enriches the training data but also reduces the risks of overfitting to the limited local data available to each client, ultimately improving performance across various tasks within the domain.

### 4.3 HETEROGENEOUS ENCODER WITH FEATURE ALIGNMENT

For the client-side computational efficiency and system scalability, we propose a heterogeneous encoder method FedDCA* to reduce the client's computational overhead by using a small encoder $\omega$ on the client side and a larger encoder $\omega'$ on the server side. However, the output dimension of heterogeneous encoders may not be consistent. Therefore, we introduce a projector $w_p$ on the server for feature alignment, as shown in Appendix A.7. We use contrastive learning (Khosla et al., 2020) and train $w_p$ on the public instructions. Specifically, $w_p$ comprises two fully connected layers and a ReLU activation layer in between, projecting $\omega$'s dimension to the output dimension of $\omega'$. For $\forall x_i \in D^p$, where $i \in \{1, 2, \ldots, \mathcal{N}\}$, $\omega$ outputs embedding $h_i$, $\omega'$ outputs embedding $h_i'$ and $w_p$ outputs embedding $\varphi_i$. For input $x_i$, the positive sample pair is $(h_i', \varphi_i)$ and negative sample pairs are $\{\varphi_j, j \neq i\}$. Let the batch size be $B$, then the training objective is defined as follows:

$$\mathcal{L} = -\sum_{i=1}^{B} \log \frac{\exp\left(\text{sim}(\varphi_i, h_i')/\tau\right)}{\sum_{j=1}^{B} \mathbb{I}_{[j \neq i]} \exp\left(\text{sim}(\varphi_i, h_j')/\tau\right)}, \tag{4}$$

where $\text{sim}(\cdot, \cdot)$ denotes the cosine similarity and $\tau$ denotes the temperature parameter.

The heterogeneous encoder setting, by employing a projector $w_p$ for feature alignment, proves effective in FL contexts where clients and the server use different encoder sizes or capacities. This approach facilitates mapping lower-dimensional client features $\mathcal{H}$ to the higher-dimensional server

space $\mathcal{H}'$ through a strategic training of $w_p$ with both positive and negative pairs. It adeptly aligns similar feature representations closely while positioning dissimilar ones further apart. As a result, the heterogeneous encoder setting not only enhances system scalability and flexibility but also facilitates the creation of a unified feature space that effectively integrates and utilizes diverse representations collaboratively.

## 4.4 Discussions

**Computation.** Through greedy client center selection, FedDCA solves the optimization problem defined in Eq. 3 in the polynomial time and is quite efficient. On the client side, the k-means algorithm (Wu, 2012) is $\mathcal{O}(|D_k^l|)$. On the server side, as mentioned in Section 4.1, the time complexity of greedy client center selection is $\mathcal{O}\left(N\left(N\xi + \xi \log \xi\right)\right)$. Specifically, for each client, the similarity computation between selected client center set $\mathcal{P}$ and client $c_k$'s cluster $\mathcal{C}_k$ is $\mathcal{O}\left(N\xi\right)$, and the sorting process is $\mathcal{O}\left(\xi \log \xi\right)$. For domain data retrieval, given that the sorting algorithm is $\mathcal{O}\left(\mathcal{N} \log \mathcal{N}\right)$, where $\mathcal{N}$ is the size of the public dataset. Thus the time complexity is $\mathcal{O}\left(N\left(\mathcal{N} \log \mathcal{N}\right)\right)$. In addition, the heterogeneous encoder setting FedDCA$^*$ further reduces the computation overhead on the client side.

**Communication.** In the domain instruction augmentation stage, each client sends $\xi$ cluster centers to the server. Next, the server performs greedy client center selection and domain data retrieval, then sends retrieved public data $D_k^g$ to the client $c_k$, whose size is $N_k^p$. In addition, in the fine-tuning stage, FedDCA follows the standard FedIT procedure, which exchanges the LoRA parameters $\Delta\phi_k$ between clients and the server. Specifically, for the Llama3-8B model, the number of trainable LoRA parameters is just 13.6 million. Compared with the number of frozen pre-trained LLM parameters, the communication for LoRA tuning is quite efficient.

**Privacy.** Comparing FedDCA with other FedIT methods (Zhang et al., 2024b; Ye et al., 2024b), the difference lies in the greedy client center selection. In this stage, the client only uploads the cluster center to the server, which is the average of embeddings to its cluster. In addition, the potential privacy leakage can be further avoided through homomorphic encryption (Acar et al., 2018), which allows the server to directly compute on ciphertext for matrix multiplication for dense retrieval. For the further concern of the domain inference attack from the server, please refer to Appendix A.6.

## 5 Experiments

To demonstrate the effectiveness of FedDCA and its variant FedDCA$^*$, we conduct extensive experiments across various domains and with several baselines. In Table 2, we highlight five critical aspects of our approach compared with other methods (Xia et al., 2024; Wang et al., 2022; Zhang et al., 2024b). For additional details and results, please refer to Appendix A.

Table 2: Key differences between FedDCA and other baselines. FedDCA demonstrates the ability of: 1) privacy preserving, 2) no API cost, 3) no additional information required, 4) avoiding performance degradation, and 5) aiming at domain coverage optimization. LESS requires additional information, as it needs access to the validation set for gradient-based retrieval.

| Method | Privacy Preserving | API Cost | Additional Information | Performance Degradation | Domain Coverage Oriented |
|---|---|---|---|---|---|
| FedAvg (McMahan et al., 2017) | ✔ | ✗ | ✗ | ✔ | ✗ |
| Random Sampling | ✔ | ✗ | ✗ | ✔ | ✗ |
| LESS (Xia et al., 2024) | ✔ | ✗ | ✔ | ✔ | ✗ |
| Self-Instruct (Wang et al., 2022) | ✗ | ✔ | ✗ | ✗ | ✗ |
| Direct Retrieval | ✔ | ✗ | ✗ | ✗ | ✗ |
| **FedDCA (ours)** | ✔ | ✗ | ✗ | ✗ | ✔ |

## 5.1 Experimental Setup

**Dataset and Evaluation Metrics.** To evaluate the performance of FedDCA, we conduct experiments on four domains: code, medical, financial, and mathematical. The detail of constructing the public dataset is described in Appendix A.5. As default, we set the number of clients to 10,

while each client has 100 local instructions and obtains 5000 augmented public instructions from the server. We analysis the effect of different retrieval amounts from 100 to 5000 on FedDCA in Appendix A.10. For evaluation, we select a range of datasets, including HumanEval (H-Eval for short) for coding (Chen et al., 2021), MMLU-Med (abbreviated as M-Med) for medical (Hendrycks et al., 2021), and GSM8K for mathematics (Cobbe et al., 2021). Additionally, financial datasets such as FPB, FiQA, and TFNS (Yang et al., 2023a) are utilized. HumanEval is evaluated using Pass@1. For MMLU-Med, FPB, FiQA, and TFNS, we use accuracy as the evaluation metric. While GSM8K is evaluated using exact match (EM). Please see Appendix A.5 for more dataset details.

**Baselines.** We compare FedDCA and FedDCA$^*$ with the following baselines: 1) Unaugmented methods, including zero-shot inference and `FedIT`, which are composed of four widely used FL methods, including FedAvg (McMahan et al., 2017), FedProx (Li et al., 2020), SCAFFOLD (Karimireddy et al., 2020) and FedAvgM (Hsu et al., 2019). 2) Augmented methods, which include random sampling, direct retrieval, LESS (Xia et al., 2024), and Self-Instruct (Wang et al., 2022). Additionally, we report FedDCA's performance with different FL strategies in Table 3. Specifically, zero-shot inference shows the performance of the pre-trained LLM without FedDIT, which gives the lower performance bound. Direct retrieval is described in Appendix A.4. Self-Instruct augments instruction in a generative way through GPT-3.5-turbo (Sun et al., 2023). Based on the prompt provided by Self-Instruct, we define the prompts for generating instructions and responses as shown in Appendix A.8.

Table 3: Performance (%) of FedDCA, FedDCA$^*$ and other nine baselines on various domains. For augmented methods, we use FedAvg as the default FL strategy. We report FedDCA with different FL strategies in the last four rows, and FedDCA+FedAvg and FedDCA+FedProx performs better across four domains.

| Method | Code | Medical | Financial | | | Mathematical |
|---|---|---|---|---|---|---|
| | H-Eval | M-Med | FPB | FiQA | TFNS | GSM8K |
| *Unaugmented Methods* | | | | | | |
| Zero-shot | 29.88 | 70.60 | 55.94 | 18.54 | 59.21 | 23.27 |
| FedAvg (McMahan et al., 2017) | 39.03 | 68.40 | 58.25 | 14.18 | 66.62 | 47.46 |
| FedProx (Li et al., 2020) | 37.20 | 69.10 | 56.51 | 14.90 | 66.45 | 47.15 |
| SCAFFOLD (Karimireddy et al., 2020) | 37.80 | 70.20 | 62.71 | 15.27 | 66.49 | 49.27 |
| FedAvgM (Hsu et al., 2019) | 32.32 | 64.70 | 68.14 | 29.27 | 70.32 | 46.85 |
| *Augmented Methods* | | | | | | |
| Random Sampling | 32.93 | 71.30 | 64.19 | 13.09 | 65.53 | 47.38 |
| Direct Retrieval | 34.14 | 72.20 | 66.31 | 19.11 | 67.62 | 50.87 |
| LESS (Xia et al., 2024) | 28.04 | 71.00 | 60.56 | 16.00 | 61.14 | 43.13 |
| Self-Instruct (Wang et al., 2022) | 32.92 | 71.90 | 59.73 | 20.67 | 66.54 | 50.79 |
| **FedDCA$^*$ (ours)** | 34.75 | 73.30 | 67.10 | 30.54 | 71.01 | 51.55 |
| *FedDCA (ours) with different FL strategies* | | | | | | |
| **FedDCA+FedAvg** | 36.58 | **74.50** | 67.24 | 35.27 | 73.32 | **52.46** |
| **FedDCA+FedProx** | 32.92 | 72.40 | **72.93** | **38.18** | **77.55** | 51.25 |
| **FedDCA+SCAFFOLD** | **39.87** | 73.20 | 72.68 | 33.09 | 75.50 | 50.26 |
| **FedDCA+FedAvgM** | 33.53 | 68.90 | 71.45 | 31.45 | 72.52 | 49.76 |

## 5.2 PERFORMANCE ANALYSIS

**Performance.** We evaluate the performance of FedDCA and compare it with several baselines on four domains (see training details in Appendix A.7). As shown in Table 3, FedDCA outperforms the other nine baselines in all domains, with a substantial improvement from at least 0.84% to the maximum of 29.19% over other baselines. In particular, FedDCA+FedAvg and FedDCA+FedProx performs better across four domains. In addition, to reduce the computation overhead of clients, FedDCA$^*$ attempts to utilize heterogeneous encoders. We see that although FedDCA$^*$ has a performance drop than FedDCA, it still outperforms other baselines. Furthermore, we report FedDCA's performance with different FL strategies, and we can see that no FL method can keep the leading position in all domains. In specific, FedProx and SCAFFOLD FL strategies perform better in the average performance. Overall, the result shows the effectiveness of FedDCA and FedDCA$^*$.

Additionally, Table 3 can be divided into two parts: unaugmented methods and augmented methods. We can observe that `FedIT` methods perform well in the code domain, even better than most augmented methods. This could be attributed to two reasons: 1) The code domain is more about following a certain paradigm. 2) As the local data is few, so compared with the same epoch and batch size, the unaugmented methods learn the same data more times, which is kind of not a fair setting. However, FedDCA+SCAFFOLD still surpasses the best baseline in the code domain, further demonstrating the effectiveness of FedDCA.

**Efficiency.** We see that direct retrieval is the best baseline in the average performance. However, it does not consider cross-client domain coverage, resulting in an overlapping retrieved data distribution. Self-Instruct (Wang et al., 2022) represents the upper limit performance of the generation-based method, which is restricted to the high expense of cost, limited API call frequency, and heavy quality screening process. On the other hand, FewFedPIT (Zhang et al., 2024b) attempts to augment local data by leveraging the pre-trained Llama2 model to perform self-instructed generation during training. However, two concerns exist: 1) The pre-trained LLM cannot provide effective, stable, and high-quality instruction generation. 2) Generating instructions locally is very costly regarding both time and computational resources. Overall, generating instructions self-instructively is not a satisfactory method for the FedDIT scenario. LESS (Xia et al., 2024) represents the augmentation methods through gradient feature retrieval. However, its performance is underwhelming. Compounding this issue, LESS presupposes access to the validation set, which is not always available, and necessitates an initial warmup on the public dataset before calculating gradients for both the public and validation data. This process becomes computationally burdensome with large public datasets. In conclusion, FedDCA achieves better performance while requiring lower computational resources and a more relaxed training condition, which shows its efficiency.

**Domain coverage.** Each method's domain coverage is shown in Table 4. Additionally, visualization of domain coverage across different strategies can be found in Appendix A.11. For augmented methods, a correlation is evident between higher domain coverage in a specific domain and improved performance of the method within that domain. Given that FedDCA aims to maximize the domain coverage through greedy client center selection, therefore it achieves the highest domain coverage in all domains, which surpasses other baselines from 4.82% to 21.36% average relative improvement. Furthermore, while FedDCA$^*$ employs a smaller encoder on the client side to enhance computational efficiency, this leads to a slight compromise in semantic precision, observed as a relative drop of 2.89% in average domain coverage. Nevertheless, FedDCA$^*$ still outperforms the best baseline by an average of 2.78%. In brief, the result proves the effectiveness of FedDCA and FedDCA$^*$ in domain coverage augmentation.

Table 4: Domain coverage of FedDCA, FedDCA$^*$ and other baselines on four domains. Since unaugmented methods do not affect domain coverage, we summarize the unaugmented methods here as `FedIT`, including four orthodox FL techniques: FedAvg, FedProx, SCAFFOLD, and FedAvgM.

| Method | Code | Med. | Fin. | Math. |
|---|---|---|---|---|
| FedIT | 0.8126 | 0.6990 | 0.8529 | 0.7871 |
| Random | 0.8512 | 0.7940 | 0.9196 | 0.8651 |
| Direct Re. | 0.9396 | 0.8830 | 0.9293 | 0.8967 |
| LESS | 0.8509 | 0.7737 | 0.8917 | 0.8352 |
| Self-Instruct | 0.8966 | 0.8586 | 0.9015 | 0.8811 |
| FedDCA$^*$ | 0.9532 | 0.8972 | 0.9538 | 0.9096 |
| FedDCA | **0.9766** | **0.9348** | **0.9815** | **0.9320** |

### 5.3 COMPUTATION ANALYSIS

Heterogenous encoder setting FedDCA$^*$ allows the system to be more scalable and flexible based on different client capacities. To show the computational efficiency of FedDCA$^*$ compared with FedDCA, we compare these two methods from the following aspects: 1) Encoder model size. 2) Encoding time overhead. We first give the evaluation setup of computation analysis.

**Evaluation setup.** The encoders used for FedDCA and FedDCA$^*$ on the client side are `bge-large-en-v1.5` and `all-MiniLM-L6-v2` respectively (detailed in Appendix A.7). Then, we show the model size and the time

Table 5: Computational cost comparison between FedDCA and FedDCA$^*$. We report the encoding time on the public dataset and the model size of each encoder.

| Method | Encoding Time | Model Size |
|---|---|---|
| FedDCA | 15 min 46 s | 335 M |
| FedDCA$^*$ | 5 min 02 s | 22.7 M |

overhead of encoding the public instructions in Table 5. Compared to FedDCA, FedDCA* has significant computational and time advantages while maintaining acceptable performance (shown in Table 3). This is accomplished by employing heterogeneous encoders and configuring a projector on the server to achieve dimensional mapping.

### 5.4 PRIVACY ANALYSIS

Next, we evaluate the privacy-preserving capability of different ratios of public data against memory extraction attacks (Carlini et al., 2021; Zhang et al., 2024a), which utilizes the autoregression nature of LLM to extract information from the memory (Xu et al., 2024).

**Evaluation setup.** We focus on one client's instruction tuning in FedDIT, using FedDCA for instruction augmentation with 1000 to 5000 public instructions. We set up 10 clients with full participation for 10 rounds. Specifically, we record the average ROUGE-L score (Lin, 2004) of client $c_0$ for each round. The designed prompt to extract instructions memorized by the LLM is detailed in Appendix A.9. For each setting, we repeat memory extraction 100 times and report the average ROUGE-L score based on each generated instruction and all local data instructions to evaluate the privacy-preserving capability of different public data amounts. Specifically, denoting the generated $\mathcal{N}$ instructions as $\mathcal{I}$ and the client's local instructions $\mathcal{I}^l$. The calculation is defined as $\frac{1}{\mathcal{N}} \sum_{i=1}^{\mathcal{N}} \texttt{ROUGE-L}(\mathcal{I}_i, \mathcal{I}^l)$.

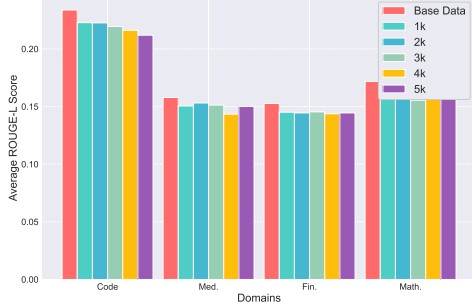 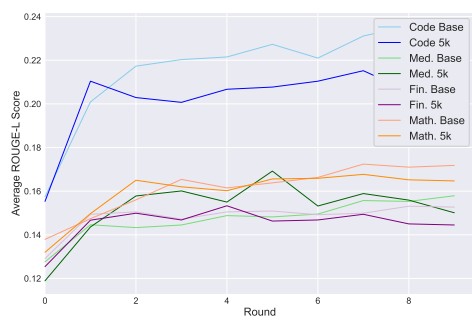

(a) Average ROUGE-L scores with different amounts of public data.

(b) Average ROUGE-L scores per round for base-data-only and augmented settings.

Figure 3: Privacy preservation analysis against memory extraction attack across four domains.

**Results.** We report the average ROUGE-L scores for base-data-only and various public data fine-tuning in Figure 3(a). The results show no significant correlation between the public data ratio and privacy-preserving capability in the same training round. In addition, only using local data has a higher risk of privacy leakage than augmented methods.

Figure 3(b) shows the average ROUGE-L score trends per round for each domain, where augmented fine-tuning uses 5000 public data. Initially, the ROUGE-L scores for augmented settings increase, then decrease or converge, while the base-data-only scores continue to rise, especially in the code domain. This indicates that with more training rounds, base-data-only fine-tuning captures more privacy information, while the privacy leakage risk in augmented fine-tuning decreases or converges.

## 6 CONCLUSION

We reveal that in the context of FedDIT, there exists a non-monotonic relationship between data heterogeneity and model performance, while the cross-client domain coverage has a significant impact on model effectiveness. In response, we propose a novel FedDIT method called FedDCA, which optimizes the domain coverage through greedy client center selection and retrieval-based instruction augmentation. Additionally, FedDCA* leverages heterogeneous encoders to reduce the client-side computation overhead and improve system scalability. Experiments across four domains demonstrate the effectiveness of FedDCA and the efficiency of FedDCA*. Further privacy analysis indicates that as fine-tuning advances, the risk of private data leakage diminishes or converges in FedDIT with instruction augmentation.

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

# A APPENDIX

## A.1 RELATED WORK

**Federated Instruction Tuning.** Instruction tuning has been widely applied across various application areas of large language models (LLM), serving as a key technique to enhance the capabilities and controllability of LLM (Zhang et al., 2023d; Wei et al., 2022). Recently, federated instruction tuning (FedIT) has emerged as an effective strategy for the distributed optimization of LLMs, leveraging federated learning (FL) protocols to improve the handling of privacy-sensitive tasks in real-world scenarios. So far, several FedIT frameworks (Ye et al., 2024b;a; Zhang et al., 2023c) have been established to evaluate the effectiveness of FedIT across multiple datasets, tasks, and FL methods. While these platforms provide a foundation for research, they have not yet introduced more complex federated algorithms and deeply investigate the challenging problems and factors affecting FedIT, which are crucial for advancing this field.

PrivateLoRA (Wang et al., 2023b) addresses privacy and efficiency issues by exploiting the low-rank properties of residual activations to reduce communication costs, significantly lowering the communication overhead through collaborative computation between the server and clients while effectively maintaining the privacy of local data. While FewFedPIT (Zhang et al., 2024b) focuses on the few-shot learning setting in FedIT, using self-generated data by pre-trained LLMs locally to mitigate the paucity of data and first discussing memory extraction attacks within FedIT.

**Domain Instruction Augmentation.** In the real world, there is an urgent need for training LLMs with specific functionalities (e.g., reasoning capabilities) or domain-specific LLMs (e.g., code (Nijkamp et al., 2023; Luo et al., 2024), medical (Zhang et al., 2023e), financial (Yang et al., 2023b;a; Zhang et al., 2023a; Wu et al., 2023), mathematical (Yue et al., 2024; Luo et al., 2023)). Existing works tend to use open-source domain-specific instruction tuning datasets for training. However, the target domain may not always have corresponding ready-made domain-specific instruction datasets. Even if they exist, these datasets are often limited in scale.

Several studies investigate domain-specific instruction augmentation, which can be categorized into three aspects: 1) Reusing human-curated public datasets (Wang et al., 2023a; Zhang et al., 2023f; Jiao et al., 2023; Lee et al., 2024; Xia et al., 2024). For instance, Parrot (Jiao et al., 2023) enhances translation capabilities of LLMs in chat by converting bilingual sentences into instruction-following datasets. Furthermore, works like INSTA (Lee et al., 2024) and LESS (Xia et al., 2024) attempt efficient domain-specific instruction augmentation via dense retrieval. INSTA (Lee et al., 2024) uses instructions without responses for effective retrieval. LESS (Xia et al., 2024) assumes access to a validation set and uses the warmup LLM's gradients of the train and validation sets for retrieval. 2) Using seed tasks for self-instruct (Luo et al., 2024; Wan et al., 2023): Explore-Instruct (Wan et al., 2023), for example, employs activate exploration to tree-model domain tasks from both depth and breadth, increasing seed instructions' coverage of the domain, and subsequently generating broader in-domain instructions through self-instruct. 3) Scaling instructions from the web: Recent works (Yue et al., 2024; Zhou et al., 2024) highlight the immense potential of mining naturally occurring instructions from the internet. Compared to generated data, web-mined instructions exhibit less bias and greater diversity. MAmmoTH2 (Yue et al., 2024) firstly retrieves domain-relevant texts and employs a LLM to extract Q-A pairs, further refining them into instruction-response pairs. Jiuzhang3.0 (Zhou et al., 2024) distills GPT's instruction generation capabilities into a smaller model and then uses it to generate instructions from the internet. In conclusion, models fine-tuned with augmented instructions have shown promising domain-specific capabilities.

To obtain a well-performing LLM in the specific domain within the distributed environment, we utilize a multi-domain dataset as public data on the server side and perform domain-specific instruction augmentation based on the client's local instructions.

## A.2 THE SETTING OF FEDDIT

### A.2.1 WHY THIS SETTING?

FedDIT enriches the local data through various instruction augmentation strategies (Zhang et al., 2024b; Xia et al., 2024; Wang et al., 2022), which can be divided into two categories: 1) Generative methods, which generate instructions through the pre-trained LLM locally or API. 2) Retrieval-based

methods, which retrieve instructions from the web. The former methods are either compromising privacy or high computational overhead. Thus, in this work, we focus on retrieval-based methods to utilize the diverse and high-quality public data.

In addition, we abstract the public data as a server-hosted multi-domain dataset. The presence of a public dataset on the server is only one possible scenario. The core innovation of this work lies in maximizing domain coverage, and the proposed algorithm is independent of the presence of a public dataset on the server. Even if the server does not have a public dataset, clients can retrieve public instructions based on the received client centers from the server, thereby achieving data augmentation that maximizes domain coverage.

Overall, this setting is chosen for simplification, allowing more attention on the federated instruction augmentation algorithms.

### A.2.2 THE DEFINITION OF DOMAIN

The concept of "domain" is flexible and hierarchical. Currently, there is no precise definition of a domain. For example, in Explore-Instruct (Wan et al., 2023), "Brainstorming" and "Rewriting" are considered two domains, but they could also be regarded as two tasks under the general domain. For clarity, we adopt a clearer domain classification in this paper (e.g., code, medical, financial, math).

### A.2.3 THE DISTRIBUTION OF PUBLIC DATA

If distributed clients aim to solve tasks based on existing knowledge, the public dataset will inevitably contain knowledge relevant to those domains. This could come from the original corpus (which can be converted into instruction-response pairs using GPT) or from pre-constructed instruction datasets on the website. So the distribution of public dataset can be categorized as follows: containing held-in or held-out instructions. The held-in indicate that the public dataset contains instructions of the specific task that clients aim to solve, while the held-out indicate that the public dataset does not contain this task's instructions.

The default setting in this work is that the public dataset contains held-in instructions. Further, we show the effectiveness of FedDCA on the held-out settings in Appendix A.10.

## A.3 WHAT TRULY COUNTS IN FEDDIT

This section will first analyze the correlation between different non-iid levels and the model's performance with separate experiments for both single and multiple domains in Section A.3.1. Further, to demonstrate the impact of non-iid on FedDIT, we compare the performance of the global model trained on augmented data based on iid and non-iid cross-client data distribution. Additionally, we suggest that domain coverage is a key factor for FedDIT in Section A.3.2.

### A.3.1 DATA HETEROGENEITY IS NOT MATTER IN FEDDIT

Following the traditional approach of constructing different degrees of non-iid, which are widely used in federated learning (Wang et al., 2020; Yurochkin et al., 2019), we adopt Dirichlet distribution to construct various heterogeneity and use k-means with the cluster num $\xi = 100$ to pseudo labeling instructions. Dirichlet distribution is affected by the hyperparameter $\alpha$, which enhances heterogeneity with a smaller $\alpha$ and decreases heterogeneity with a larger $\alpha$. We choose four widely used heterogeneity, which are $\alpha = [0.01, 0.1, 1, 10]$ (Ye et al., 2023; Zhang et al., 2023b; Li et al., 2021). Figure 4 shows a visualization of the data distribution with different heterogeneity on the code dataset.

We perform instruction tuning on different amount of clients with different heterogeneity. Specifically, the client's number are 10 and 100 respectively with 2 randomly selected clients participate in each round. For each domain, we only use the in-domain data and then perform FedIT. The training details are shown in Appendix A.7. We perform the experiments 3 times with different random seeds (42, 43 and 44) and report the average performance and the standard deviation in each domain. As shown in Table 6, the performance of LLM does not decrease due to the increase of data heterogeneity but shows a non-monotonic correlation, which indicates that the performance of LLM does not directly depend on data heterogeneity and other factors that play a key role.

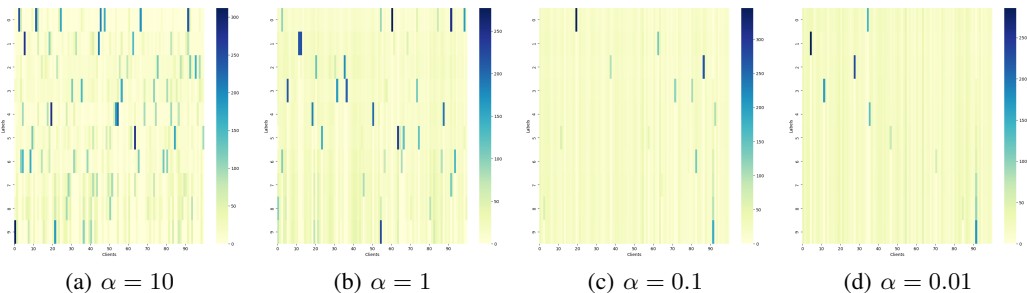

|       | (a) $\alpha = 10$ | (b) $\alpha = 1$ | (c) $\alpha = 0.1$ | (d) $\alpha = 0.01$ |

Figure 4: Visualization of client data distribution with $\alpha = [10, 1, 0.1, 0.01]$ in the code domain.

| #Clients | Metric | $\alpha = 10$ | $\alpha = 1$ | $\alpha = 0.1$ | $\alpha = 0.01$ |
|---|---|---|---|---|---|
| 10 | H-Eval | $36.17 \pm 1.03$ | $34.54 \pm 0.35$ | $32.71 \pm 2.75$ | $34.75 \pm 2.65$ |
|  | M-Med | $71.80 \pm 0.84$ | $71.60 \pm 0.70$ | $71.33 \pm 0.37$ | $71.56 \pm 1.37$ |
|  | FPB | $66.41 \pm 3.31$ | $68.72 \pm 4.54$ | $68.39 \pm 2.65$ | $70.07 \pm 3.50$ |
|  | FiQA | $22.90 \pm 9.34$ | $32.48 \pm 9.85$ | $26.42 \pm 6.67$ | $38.42 \pm 8.78$ |
|  | TFNS | $69.45 \pm 2.34$ | $72.99 \pm 3.20$ | $71.65 \pm 1.64$ | $73.66 \pm 2.90$ |
|  | GSM8K | $56.51 \pm 0.14$ | $59.28 \pm 0.38$ | $56.75 \pm 0.23$ | $57.64 \pm 0.29$ |
| 100 | H-Eval | $36.57 \pm 1.84$ | $36.57 \pm 3.48$ | $34.12 \pm 0.26$ | $36.88 \pm 0.13$ |
|  | M-Med | $70.73 \pm 0.37$ | $72.20 \pm 0.45$ | $71.83 \pm 0.37$ | $71.60 \pm 0.45$ |
|  | FPB | $67.87 \pm 1.13$ | $64.10 \pm 2.07$ | $66.71 \pm 1.97$ | $67.59 \pm 1.42$ |
|  | FiQA | $33.44 \pm 3.01$ | $19.87 \pm 5.83$ | $33.93 \pm 3.54$ | $33.69 \pm 6.58$ |
|  | TFNS | $72.22 \pm 0.74$ | $70.13 \pm 2.59$ | $73.01 \pm 0.22$ | $71.99 \pm 0.95$ |
|  | GSM8K | $54.32 \pm 0.41$ | $51.27 \pm 0.28$ | $58.94 \pm 0.36$ | $57.81 \pm 0.19$ |

Table 6: Performance (%) of different heterogeneity in each domain with 10 and 100 clients.

### A.3.2 DOMAIN COVERAGE: A KEY FACTOR IN FEDDIT

Explore-Instruct (Wan et al., 2023) enhances the coverage of domain-specific seed tasks through active exploration, then uses the self-instruct method for instruction data augmentation. This approach highlights the impact of domain coverage on domain-specific instruction tuning. Inspired by Explore-Instruct, we attempt to conduct more in-depth and extensive experiments to study the effect of domain coverage on FedDIT. Firstly, we define the domain coverage of cross-client data in the FL setting. Assume the dataset of in-domain data $D^d$ represents the latent data distribution of this domain and the cross-client data is defined as $D^c = \cup_{k=1}^{N} \left( D_k^l \cup D_k^g \right)$. Inspired by the facility location function (Cornuejols et al., 1983), we define the domain coverage of $D^c$ respect to $D^d$ as follows:

$$d(D^d, D^c) = \frac{1}{|D^d|} \sum_{d \in D^d} \max_{v \in (D^c \cap D^d)} sim(d, v), \qquad (5)$$

where $sim(d, v)$ is the similarity between $d$ and $v$. Specifically, we use the cosine similarity function in FedDCA. Note that we only use the in-domain data in $D^c$ to calculate the domain coverage because the out-of-domain data would mislead the domain coverage evaluation, as shown in Figure 5.

To better align with the real-world scenarios, we explore instruction augmentation based on both iid and non-iid cross-client data distribution and adopt direct data retrieval as described in Appendix A.4 for FedDIT. We set the number of clients to 10, while each client has 100 local instructions and obtains 5000 augmented public instructions from the server.

Table 7: Performance(%) and domain coverage of iid and non-iid settings on different domains. The higher domain coverage correlates with better performance.

| Test Set | Performance (%) | | Domain Coverage | |
|---|---|---|---|---|
|  | iid | non-iid | iid | non-iid |
| H-Eval | **35.36** | 33.53 | **0.8538** | 0.7994 |
| M-Med | 70.20 | **71.00** | 0.7800 | **0.8027** |
| FPB | 58.58 | **64.19** |  |  |
| FiQA | 17.09 | **19.27** | 0.8523 | **0.9327** |
| TFNS | 66.16 | **69.09** |  |  |
| GSM8K | **40.50** | 38.50 | **0.9137** | 0.8448 |

To construct iid data distribution, we randomly sample 1,000 from multi-domain datasets (code, medical, financial, mathematical, and general), which is detailed in Table 8 and divide them into 10 shards as each client's local data. To construct non-iid data distribution, we perform k-means clustering with $\xi = 100$. Each client randomly samples 100 instructions from different randomly selected clusters. Furthermore, for both iid and non-iid settings, direct data retrieval is performed based on the client's local data.

Table 7 presents the performance of FedDIT on different domains with iid and non-iid settings and shows the domain coverage in four domains. We can observe that both iid and non-iid settings outperform in some domains, but both collectively indicate that higher domain coverage correlates with better performance.

○ : In-domain data
△ : Out-of-domain data

Figure 5: Misleadning of out-of-domain data on domain coverage calculation.

### A.4 DIRECT DOMAIN DATA RETRIEVAL

In this section, we first show the detail of instruction-based dense retrieval in Appendix A.4.1, which is both used in direct retrieval and FedDCA. Then, we explain the direct retrieval algorithm in Appendix A.4.2.

#### A.4.1 INSTRUCTION BASED DENSE RETRIEVAL

Suppose an instruction dataset $\mathcal{D}$ consists of several instances. Each instance is a (*Instruction*, *Response*) pair. For instructions that have *Input*, we concatenate the *Instruction* and *Input* as *Instruction*, which is consistent with OpenFedLLM (Ye et al., 2024b). Then we use only the instruction for encoding and dense retrieval. Denote $\mathcal{E}$ as a cluster center and $I$ as an instruction of the public dataset $D^p$, then we measure the instruction-based similarity score for dense retrieval as follows: **Score**$(\mathcal{E}, I) = sim(\mathcal{E}, w_{enc}(I))$, where $sim(\cdot, \cdot)$ is the cosine similarity function. Based on the computed similarity between $\mathcal{E}$ and each $I$ in the public data, we then select the top-$N_k^p$ instructions as the retrieved public data for domain data augmentation.

---

**Algorithm 2** Direct domain data retrieval

**Parameters:**
Clients' local datasets $D = \{D_1, D_2, \ldots, D_N\}$; Public dataset $D^p$; Local datasets $D^l = \{D_1^l, D_2^l, \ldots, D_N^l\}$; Number of clusters $\xi$; Encoder model $w_{enc}$; Pre-encoded public instruction embeddings $\mathcal{E}$.

1: $D^g \leftarrow \emptyset$
2: **for** $i \in \{1, 2, \ldots, N\}$ **do**
3:      $\mathcal{E}' \leftarrow \{w_{enc}(x) \mid x \in D_{i,instruction}^l\}$            ▷ Encode the local instructions
4:      $\mathcal{C} \leftarrow$ k-means$(\xi).fit(\mathcal{E}')$       ▷ Cluster local instructions, return cluster centers $\mathcal{C}$
5:      $\mathcal{S} \leftarrow \mathcal{C} \cdot \mathcal{E}^T$                 ▷ Compute the similarity score between $\mathcal{C}$ and $\mathcal{E}$
6:      $\mathcal{I} \leftarrow \emptyset$
7:      **for** $j \in \{1, 2, \ldots, \xi\}$ **do**
8:          $\mathcal{S}' \leftarrow \{s \mid s \in S_j, \text{index}(s) \notin \mathcal{I}\}$       ▷ Filter the selected indices
9:          $\mathcal{I}' \leftarrow$ indices of the top-$\frac{N_k^p}{\xi}$ elements in $\mathcal{S}'$     ▷ Retrieve the top $\frac{N_k^p}{\xi}$ indices
10:          $\mathcal{I} \leftarrow \mathcal{I} \cup \mathcal{I}'$               ▷ Update the selected indices
11:      **end for**
12:      $D_i^g \leftarrow \{D_j^p \mid j \in \mathcal{I}\}$            ▷ Selected public instructions
13:      $D_i \leftarrow D_i^g \cup D_i^l$         ▷ Obtain the augmented instruction dataset $D_i$
14: **end for**

---

#### A.4.2 DIRECT RETRIEVAL

The direct domain data retrieval only utilizes instructions of the client's local data without responses to perform the retrieval-based domain data augmentation. The detailed algorithm is described in

Algorithm 2. For each client, we start by encoding the local instructions using the encoder model $w_{enc}$. Next, we apply the k-means algorithm to cluster the embeddings into $\xi$ clusters. Then $\xi$ cluster centers are sent to the server for retrieval-based domain data augmentation. Subsequently, the retrieved public data is sent to the client for instruction tuning.

## A.5 TRAIN AND TEST DATASET INFORMATION

Here we provide the train and test dataset details of code (CodeAlpaca[2]), medical (MedAlpaca[3]), financial (FinGPT (Yang et al., 2023a)) and mathematical (MathInstruct[4]) domain, respectively.

Table 8: Dataset information of each domain.

| Dataset name | Domain | Type | $N_{sample}$ | Metric |
|---|---|---|---|---|
| CodeAlpaca | Code | Train | 20,022 | - |
| HumanEval | Code | Test | 164 | Pass@1 |
| MedAlpaca | Medical | Train | 33,955 | - |
| MMLU-Med | Medical | Test | 1,089 | Acc |
| FinGPT | Financial | Train | 76,772 | - |
| FPB | Financial | Test | 152 | Acc |
| FiQA | Financial | Test | 35 | Acc |
| TFNS | Financial | Test | 299 | Acc |
| MathInstruct | Mathematical | Train | 224,567 | - |
| GSM8K | Mathematical | Test | 1,319 | Exact Match |
| Alpaca | General | Train | 52,002 | - |

As shown in Table 8, the public data consists of four domain-specific instruction datasets and a general instruction dataset, which are CodeAlpaca, MedAlpaca, FinGPT, MathInstruct, and Alpaca, respectively. For each domain's FedDIT, we randomly select 1000 samples from the in-domain instruction and split them into 10 shards as each client's local dataset. The rest of the instructions are used as the public dataset $D^p$. For evaluation, we use HumanEval for the code domain, MMLU-Med for the medical domain (specifically using subjects `anatomy`, `clinical_knowledge`, `college_biology`, `college_medicine`, `medical_genetics` and `professional_medicine` in MMLU), FPB, FiQA and TFNS for the financial domain, and GSM8K for the mathematical domain.

## A.6 DISCUSSION ON DOMAIN INFERENCE ATTACK

The server may inference the clients' data domain when the domain data retrieval is performed on the server side. However, the proposed algorithm FedDCA is independent of the presence of a public dataset on the server. Even if the server does not have a public dataset, clients can upload their cluster centers to the server, which selects a set of client centers and sends them back to the clients. Each client can then retrieve data from the website based on the received client center by itself, thereby achieving data augmentation while maximizing the cross-client domain coverage.

In that case, since the server does not know the encoder used by the client, it cannot infer the semantic meaning of the embedding. Thus, for the server, it becomes significantly more challenging to infer the client's domain, let alone apply any privacy protection techniques to the embeddings.

We provide two examples for illustration. Two different encoders are used as client's and server's respectively: `BAAI/bge-large-en-v1.5` (denoted as $w_1$) and `google-bert/bert-large-uncased` (denoted as $w_2$). Both encoders output 1024-dimensional features.

---

[2] https://huggingface.co/datasets/sahil2801/CodeAlpaca-20k
[3] https://huggingface.co/datasets/medalpaca
[4] https://huggingface.co/datasets/TIGER-Lab/MathInstruct

**Example 1:** Both $w_1$ and $w_2$ take `"hello world"` as input, and the cosine similarity between their embeddings is **0.1829**.

**Example 2:** Three instructions are used:

- **Instruction 1:** Create an array of length 5 which contains all even numbers between 1 and 10.

- **Instruction 2:** Write a replace method for a string class which replaces the given string with a given set of characters.

- **Instruction 3:** What is the sentiment of this news? Please choose an answer from {negative/neutral/positive}. Teollisuuden Voima Oyj, the Finnish utility known as TVO, said it shortlisted Mitsubishi Heavy's EU-APWR model along with reactors from Areva, Toshiba Corp., GE Hitachi Nuclear Energy, and Korea Hydro & Nuclear Power Co.

For these instructions, Instruction 1 is passed to $w_1$ and Instructions 2 and 3 are passed to $w_2$, which will result in three embeddings: $e_1$, $e_2$, and $e_3$. The cosine similarity between $e_1$ and $e_2$ is **0.1464**, while the similarity between $e_1$ and $e_3$ is **0.1879**. Instructions 1 and 2 are in the same domain, whereas they have a lower cosine similarity.

In conclusion, as demonstrated above, when the clients do not perform domain-specific instruction retrieval on the server side, the server cannot infer the client's domain based on the uploaded embeddings.

### A.7 IMPLEMENTATION DETAILS

We consider FedDIT in the cross-device scenario, $N = 10$ clients, $\mathcal{R} = 30$ rounds, where we randomly sample 2 clients to be available for each round. Then, each available client performs FedDIT for 10 steps with AdamW optimizer, and the batch size is $B = 32$ in a round. The initial learning rate is $5e - 5$ with a cosine learning rate scheduler. Our experiment utilizes the widely used LLM, Llama3-8B[5] as the base model with 2048 max sequence length and adopts LoRA tuning method. The rank of LoRA is 16, and the scalar alpha is 16. For k-means (Wu, 2012), we set cluster num $\xi = 10$ and for FedDCA we set the similarity threshold $\alpha = 0.7$. For FedDCA*, we set the temperature parameter $\tau = 0.5$ for contrastive learning (Khosla et al., 2020). We utilize `bge-large-en-v1.5`[6] as both the client and server's encoder as default, which outputs embeddings of 1024 dimensions. While we utilize `all-MiniLM-L6-v2`[7] as the client's small encoder which outputs embeddings of 384 dimensions and `bge-large-en-v1.5` as the server's encoder for FedDCA*.

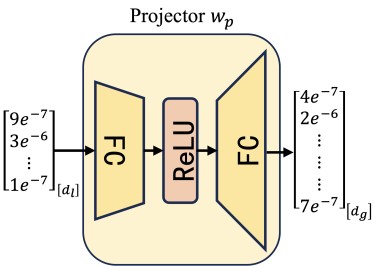

Figure 6: The projector used on the server side for feature alignment, which consists of two fully connected layers and a ReLU activation layer in between.

**Projector.** To reduce the computational overhead of clients, we propose a heterogeneous encoder method called FedDCA*. We utilize a projector to perform the dimension alignment between the small encoder on the client side and the large decoder on the server side. As shown in Figure 6, where $d_l$ is the dimension of the client-side encoder, $d_g$ is the dimension of the server-side encoder.

### A.8 PROMPTS USED IN THE SELF-INSTRUCT DATA GENERATION

To generate the Self-Instruct data, we prompt GPT-3.5 to generate the instruction with the designed prompt in Figure 7. Specifically, we randomly sample two examples from the client's local data to guide GPT-3.5 generating the in-domain instruction and one example from the client's local data for one-shot in-context learning to guide GPT-3.5 generating responses into the example's format.

---

[5] `https://huggingface.co/meta-llama/Meta-Llama-3-8B`
[6] `https://huggingface.co/BAAI/bge-large-en-v1.5`
[7] `https://huggingface.co/sentence-transformers/all-MiniLM-L6-v2`

> You are asked to come up with
> instructions. Don't repeat instructions in
> examples. Here are some examples:
> Instruction 1: {}
> Instruction 2: {}
> Provide a new instruction below:

> Example 1:
> Instruction: {}
> Response: {}
> Generate the response of this instruction,
> Instruction: {}

Figure 7: Prompts used in the Self-Instruct data generation. (a) Prompt for generating new instructions. Two examples are randomly sampled from the client's local data for in-context demonstration. (b) Prompt for generating responses. We prompt GPT-3.5 to generate responses with a randomly selected example for one-shot in-context learning.

### A.9 PROMPT USED FOR MEMORY EXTRACTION ATTACK

As we use Llama3-8B as our base model and format the instructions and responses into the Alpaca's format, to utilize the auto-regression nature of LLM to extract the instruction, we prompt the model to generate the instruction using the prompt in Figure 8, which is exactly the prefix of the Alpaca's template.

> Below is an instruction that describes a
> task. Write a response that appropriately
> completes the request.
> ###Instruction:

Figure 8: Prompt used for memory extraction attack.

### A.10 FURTHER ANALYSIS

To further study the effect of different hyperparameters of FedDCA, we undertake a thorough analysis including various retrieval amounts and different k-means cluster numbers $\xi$. In addition, we perform the ablation study on whether using the similarity threshold $\alpha$ in the greedy client center selection.

**Held-out Setting.** Considering this setting in the financial domain, to construct the held-out setting, given that the training set FinGPT and the test sets FPB, FiQA, and TFNS are all related to sentiment analysis tasks. As clients aim to train a model adept at performing sentiment analysis through federated learning, we keep the setting of test sets. Meanwhile, the FinGPT's instructions in public data are replaced with data from the `Sujet-Finance-Instruct-177k` dataset where `task_type=qa`. The clients' local data are still randomly sampled from FinGPT. This approach yields held-out public data.

Table 9: Performance (%) on the test set of financial domain after 30 rounds' federated instruction tuning.

| Method | FPB | FiQA | TFNS |
|---|---|---|---|
| Zero Shot | 55.94 | 18.54 | 59.21 |
| Base Data | 58.25 | 14.18 | 66.62 |
| Random Sampling | 60.39 | 9.45 | 65.45 |
| **FedDCA** | **60.89** | **18.91** | **67.37** |

As shown in Table 9, it can be observed that even when in the held-out setting, FedDCA still achieves performance improvements compared to other baselines. Additionally, using the Random Sampling data augmentation strategy resulted in performance degradation on the FiQA dataset. This further underscores the necessity of selecting an appropriate data augmentation strategy.

**Effect of Retrieval Number.** We report the performance and the corresponding domain coverage of FedDCA with different retrieval amounts on the four domains in Figure 9, respectively. We can see that the domain coverage of FedDCA is increasing along with the retrieval amount in different trends, as well as the performance. Specifically, the domain coverage of each domain increases by 6.36%, 18.69%, 5.04%, and 8.06% in relative, respectively. Along with domain coverage increasing, the performance of FedDCA is increasing by 3.05%, 2.20%, 4.08%, and 6.95% for each domain. Noted that although the code domain is more about following a certain paradigm, which could perform well with a few data and more fine-tuning rounds, it still could benefit from the instruction augmentation.

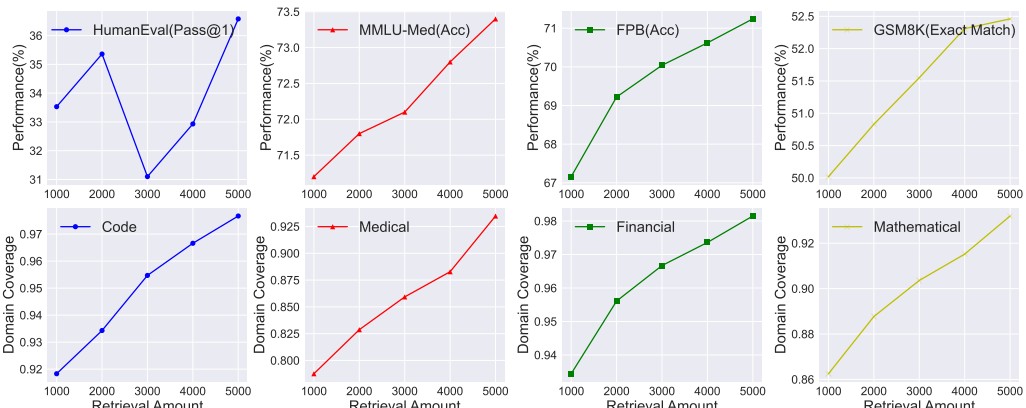

Figure 9: Effect of different retrieval amounts on the performance of FedDCA and its domain coverage. We show the results on four domains separately. Here, we use the FPB test set to evaluate the performance in the financial domain.

In addition, to further prove the effectiveness of FedDCA is independent of the retrieval amount, we conduct the experiment that each client samples 100 samples from the public dataset and then performs FedDIT. Random* and FedDCA* represent the default setting, where each client samples 5,000 samples.

Table 10: Performance (%) of FedDCA and Random Sampling with different amounts of sampled public data.

| Method | H-Eval | MMLU-Med | FPB | FiQA | TFNS | GSM8K |
|---|---|---|---|---|---|---|
| Base Data | 39.03 | 68.40 | 58.25 | 14.18 | 66.62 | 47.46 |
| Random Sampling | 34.53 | 69.80 | 60.89 | 14.54 | 65.57 | 48.77 |
| Random Sampling* | 32.93 | 71.30 | 64.19 | 13.09 | 65.53 | 47.38 |
| FedDCA | 35.97 | 70.20 | 63.20 | 15.63 | 67.58 | 49.32 |
| FedDCA* | 36.58 | 74.50 | 67.24 | 35.27 | 73.32 | 52.46 |

The results in Table 10 show that even with a small amount of sampled data, the model performance still improves compared to random sampling, except in the code domain, which tends to follow a certain paradigm. This further demonstrates the effectiveness of FedDCA and its independence of the retrieval amount.

**How Far can FedDCA Go?** Assume there exists a pre-trained model that has already fine-tuned on the whole in-domain data of the internet or even all open-source public instruction datasets, is there any room for instruction augmentation or FedDCA? To answer this question, we conduct the following experiments on four settings: A) Fine-tuning on the whole public dataset. B) Following A, perform further federated fine-tuning with only private clients' local dataset. C) Following A, each client randomly samples 100 public data for fine-tuning. D) Following A, each client samples 100 public data through FedDCA and then performs FedDIT.

As shown in Table 11, even with only 100 samples selected via FedDCA, the method not only prevents performance degradation but also helps the model achieve better generalization within the

Table 11: Performance (%) of setting A, B, C, and D in each domain.

| Setting | H-Eval | M-Med | FPB | FiQA | TFNS | GSM8K |
|---------|--------|-------|-------|-------|-------|-------|
| A | 38.41 | 69.10 | 75.33 | 41.09 | 71.60 | 53.75 |
| B | 43.90 | 64.30 | 77.97 | 66.54 | 78.26 | 56.40 |
| C | 41.46 | 66.90 | 74.50 | 29.81 | 70.51 | 57.01 |
| D | 44.12 | 70.90 | 81.43 | 72.00 | 78.81 | 58.90 |

domain. This is attributed to exposure to more diverse in-domain data during training. Meanwhile, it can be observed that random sampling still leads to performance degradation compared to Setting A, and even performs worse than Setting B, where no data augmentation is applied. In the financial domain, performance degradation is also evident.

In conclusion, this experiment further highlights the necessity and effectiveness of FedDCA and designing effective data augmentation algorithms for FedDIT.

**Scalability.** To evaluate the scalability of FedDCA, we conduct the experiment with 100 clients. In each round, two clients are randomly selected for federated instruction tuning using FedAvg.

Table 12: Performance (%) and domain coverage of FedDCA and other baselines in various domains. The client number is 100, and 2 clients are selected in each round.

| | Zero-shot | Base Data | Random Sampling | FedDCA |
|---------|-----------|-----------|-----------------|--------|
| | | *Performance (%)* | | |
| H-Eval | 29.88 | 34.14 | 34.75 | **35.92** |
| MMLU-Med | 70.60 | 72.40 | 69.90 | **73.30** |
| FPB | 55.94 | 66.74 | 61.05 | **67.16** |
| FiQA | 18.54 | 33.45 | 12.00 | **34.51** |
| TFNS | 59.21 | 72.78 | 65.53 | **73.26** |
| GSM8K | 23.27 | 49.12 | 47.23 | **50.26** |
| | | *Domain Coverage* | | |
| Code | - | 0.8282 | 0.8685 | **0.9242** |
| Med. | - | 0.8377 | 0.8497 | **0.9090** |
| Fin. | - | 0.9339 | 0.9408 | **0.9800** |
| Math. | - | 0.8709 | 0.8812 | **0.9118** |

As shown in Table 12, as the number of clients increases, the amount of local data on each client gradually grows. The model trained using only base data even outperforms random sampling in domains other than code and narrows the gap with FedDCA. However, because FedDCA aims to maximize the cross-client domain coverage, it achieves higher domain coverage and better performance. In conclusion, results further demonstrate the effectiveness and scalability of FedDCA.

**Impact of Different Cluster Number.** The hyperparameter $\xi$ is the number of clusters in the k-means algorithm. The experiment is conducted on $\xi = [N, 2N, 4N, 8N]$, where $N$ is the number of clients. Following the setting in Appendix A.7, we set $N = 10$. We report the domain coverage of the augmented dataset via FedDCA with different cluster numbers $\xi$ on the four domains in Figure 10, respectively. Results show that there is no best $\xi$ for all domains. Specifically, the best $\xi$ of code, medical, financial, and mathematical domains are 80, 80, 40, and 10, respectively.

**Ablation Study.** We conduct the experiment with FedDCA w/o similarity threshold $\alpha$ on the four domains based on the FedAvg FL strategy. The performance and the corresponding domain coverage are shown in Table 13, where FedDCA without the similarity threshold $\alpha$ is marked as FedDCA[†]. The result shows that the performance of FedDCA with similarity threshold $\alpha$ is slightly better than FedDCA without using $\alpha$ in code, financial, and mathematical domains, as the similarity scores in the medical domain are relatively lower. We show the similarity score distribution of the four domains in Figure 11. For each domain, we plot each similarity score's distribution of 10 clients.

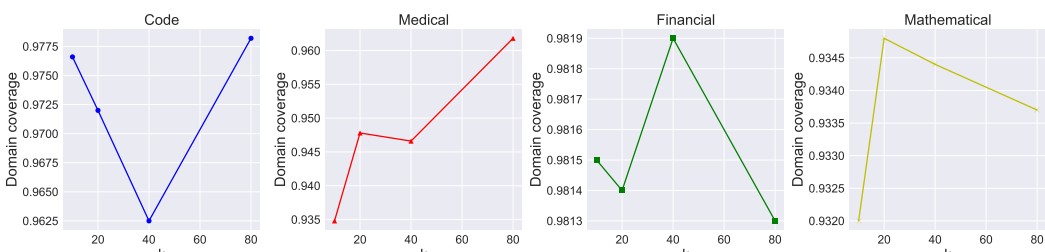

Figure 10: Impact of different cluster number $\xi$ on the cross-client domain coverage. We report the domain coverage of the augmented dataset via FedDCA with different cluster numbers $\xi$ on the four domains.

The similarity score is computed between the selected client center and the public data. Then, we show the similarity score distribution using the histogram plot.

Table 13: Ablation study on the performance and domain coverage of FedDCA w/o similarity threshold $\alpha$.

| Metric | Performance(%) | | Domain Coverage | |
|---|---|---|---|---|
| | FedDCA$^{\dagger}$ | FedDCA | FedDCA$^{\dagger}$ | FedDCA |
| H-Eval | 35.97 | **36.58** | 0.8972 | **0.9348** |
| M-Med | 73.40 | **73.40** | 0.9348 | **0.9348** |
| FPB | 66.25 | **67.24** | | |
| FiQA | 23.27 | **35.27** | 0.9353 | **0.9815** |
| TFNS | 69.34 | **73.32** | | |
| GSM8K | 51.78 | **52.46** | 0.9128 | **0.9320** |

### A.11 AUGMENTATION STRATEGY VISUALIZATION

To more intuitively compare the domain coverage of different instruction augmentation methods, we randomly sample 5,000 instructions obtained through these methods and 10,000 in-domain instructions as the background, representing the distribution of specific domains in the public dataset. We then visualized the results using t-SNE (van der Maaten & Hinton, 2008), as shown in Figure 12. The plot shows that FedDCA encompasses most of the in-domain data, which is consistent with FedDCA's domain coverage of each domain shown in Table 4. Also, we can observe that the random sampling strategy selects a lot of out-of-domain data while does not have good coverage in specific domains.

### A.12 FREQUENTLY USED NOTATION

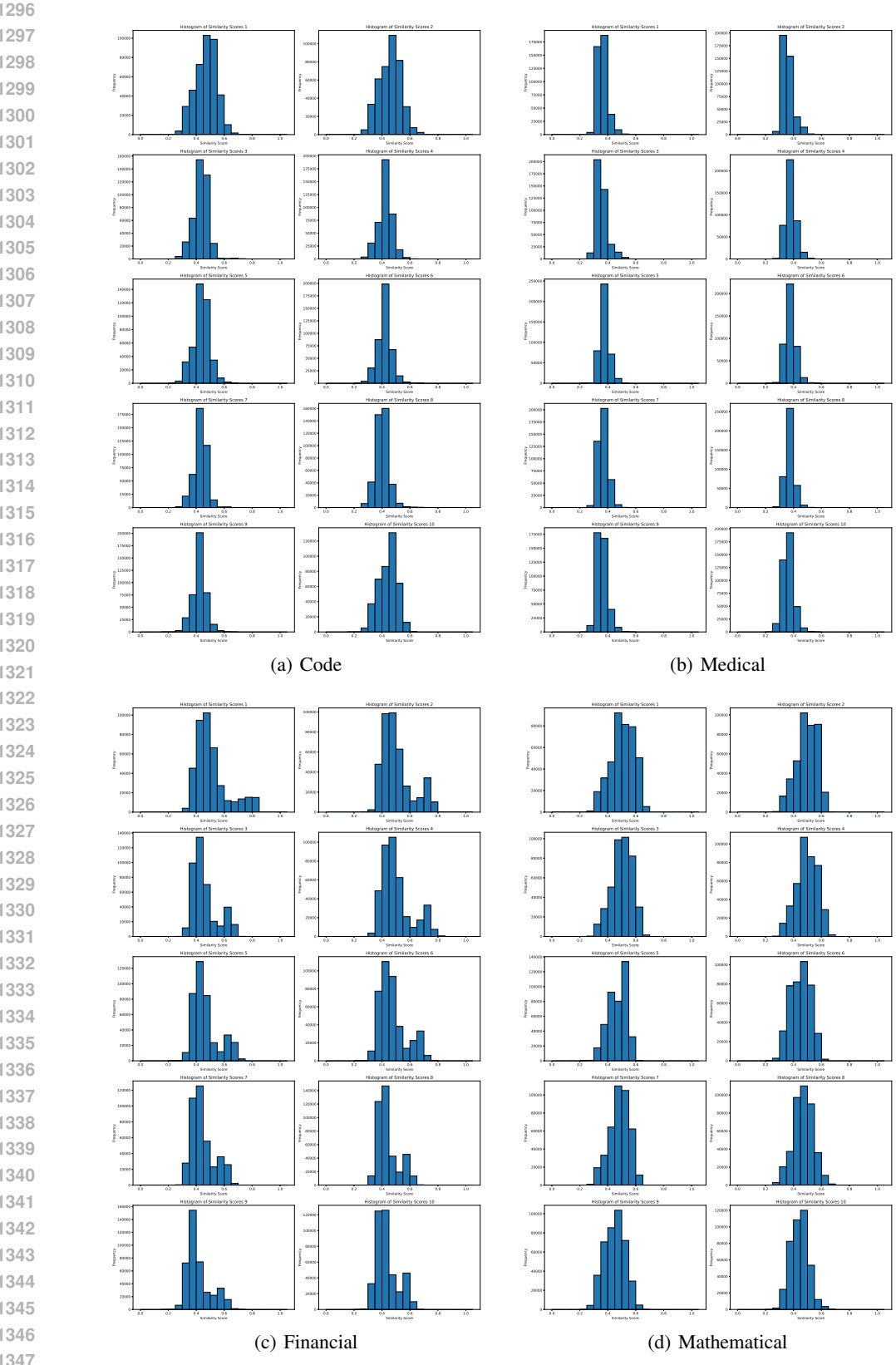

Figure 11: The similarity score distribution of the four domains. For each domain, we plot each similarity score's distribution of 10 clients.

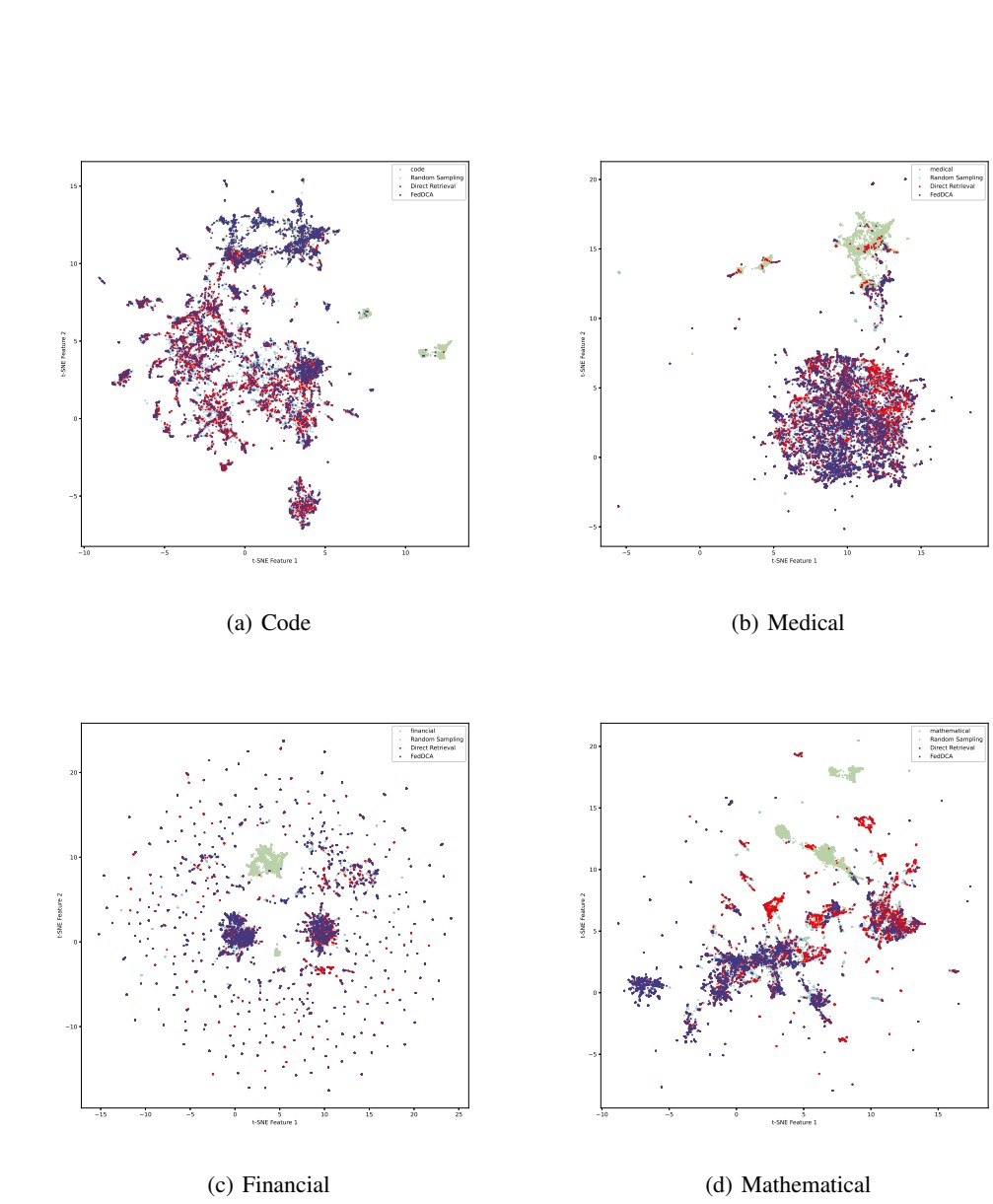

(a) Code

(b) Medical

(c) Financial

(d) Mathematical

Figure 12: Visualization of cross-client data distribution in different domains, performing t-SNE dimensionality reduction on retrieved instructions through various augmentation strategies. We randomly sample 10,000 in-domain samples as background while randomly sampling 5,000 samples from the cross-client augmented dataset for different instruction augmentation methods for comparison.

Table 14: Frequently used notation.

| NOTATIONS | REMARK |
|---|---|
| $N, C$ | Clients number, client set $C = \{c_1, c_2, \ldots, c_N\}$. |
| $D^p, D^d, D^c, D^l$ | The public datasets, the in-domain data, the cross-client dataset, client's local private data. |
| $N_k^l, N_k^p$ | Number of local private data on the $k$-th client, number of the retrieved public data on the $k$-th client. |
| $D_k^l, D_k^g, D_k$ | The local private data on the $k$-th client, the retrieved public data on the $k$-th client, the augmented dataset on the $k$-th client. |
| $\Lambda, \mathcal{P}, \mathcal{C}$ | A specific sampling strategy that performs instruction augmentation on the server side, selected client center set, the cluster centers obtained locally and sent to the server for the greedy client center selection. |
| $\phi, \Delta\phi$ | LLM's pre-trained parameters, additional LoRA parameters. |
| $w, w_{enc}, w_p$ | Merged model parameters from the frozen LLM's parameters $\phi$ and the additional LoRA parameters $\Delta\phi$, encoder model, projector model. |
| $F_k(w; \mathcal{D}), l(w; x, y)$ | Loss of model $w$ over a specific dataset $\mathcal{D}$, the instructon tuning loss of model $w$ over a data sample $(x, y)$. |

