# OpenReview forum: "Federated Instruction Tuning of LLMs with Domain Coverage Augmentation"
_ICLR.cc/2025/Conference — Submitted to ICLR 2025_

### Official Review · Reviewer_8vFk · 2024-10-27

**Soundness:** 1
**Presentation:** 1
**Contribution:** 2
**Rating:** 3
**Confidence:** 4

**Summary:**

This paper proposes a federated instruction tuning approach. The authors assume the existence of a large public dataset that covers all domain data, while each domain only has a small subset of data. The paper introduces multiple techniques, including greedy client center selection and domain data retrieval, to achieve data selection from the public dataset and improve the performance of federated instruction tuning. Furthermore, a heterogeneous encoder approach is introduced to further enhance efficiency. The experiments demonstrate that this data selection approach improves performance compared to client-data-only fine-tuning baselines.

**Strengths:**

1. This approach highlights domain coverage as the main bottleneck for federated instruction tuning.

**Weaknesses:**

1. The finding that motivates this research—"there is no linear correlation between the degree of non-independent and identically distributed (IID) and LLM performance in the context of FedDIT"—requires further evidence and rephrasing to avoid being misleading. First, "linear correlation" is a very strong term and may be inappropriate here; the authors likely mean "monotonicity" instead of "linearity." Second, the current experiments are insufficient to draw definitive conclusions. The results are not conducted over multiple rounds, nor do they report the mean and standard deviation of performance, which makes them highly dependent on the seed, especially in the Dirichlet instance split. Third, the number of clients (10, with 2 clients involved in each round) is relatively small and may not be significantly affected by \$α\$. Fourth, averaging performance across multiple datasets or tasks is problematic. For instance, in Table 1, Method 2 is clearly better than Method 1 on multiple datasets, but the average shows the opposite.

**Table 1. Performance of Method 1 and Method 2 on Binary Classification Tasks**

| Method  | Method 1  | Method 2 |
| ------- | --------- | -------- |
| Data 1  | 98%       | **99%**  |
| Data 2  | 98%       | **99%**  |
| Data 3  | 98%       | **99%**  |
| Data 4  | 98%       | **99%**  |
| Data 5  | **52%**   | 47%      |
| Average | **88.8%** | 88.6%    |

2. The assumption that there exists a public dataset requires further clarification regarding its prevalence in practice. Clarification is needed on whether there are real-world applications that align with this assumption. It is also necessary to determine how much of the improvement is due to the additional data versus the algorithmic design. Furthermore, it should be clarified whether the public multi-domain dataset needs to be within the domain of the clients, or if it can be out-of-domain data.

3. The method of retrieving similar public instructions to domain instructions requires the public dataset to already cover all domains (i.e., contain full domain knowledge). However, it is unclear which baseline uses this large, powerful public dataset for fine-tuning. It would be interesting to see the results of:

- Fine-tuning with the same public dataset.
- Fine-tuning with the public dataset followed by further federated fine-tuning with private clients' local dataset.

Including these baselines would significantly enhance the convincing nature of the experiments.

4. The experimental setting is unrealistic. In line 996, the authors randomly select 1,000 samples from the in-domain instruction set and split them into 10 shards as each client's local dataset, leaving the remaining 20x to 200x samples for the public dataset. The necessity of federated learning is questionable if such a large, independent and identically distributed (IID), noise-free public dataset exists.

**Minor Comments**

1. The format of Table 1 is difficult to read. It is unclear which entries represent methods and which represent metrics, as well as how to interpret the different metrics.

**Questions:**

1. Does random sampling baseline make a sample of each client's domain data? In Table 1, random sampling generally outperforms full-data fine-tuning. However, the introduction claims that FedIT requires a sufficient amount of instruction data. If this is true, why does down-sampling lead to an increase in average performance? More evidence is needed to support the motivation.

2. In "Results demonstrate that domain coverage significantly influences model performance in the corresponding domain," is this conclusion drawn from the experiments in this manuscript or from Wang et al. (2023)?

---

> ### Author Response · Authors · 2024-11-20
>
> Thank you for your time and valuable suggestions. Here are our detailed responses.
>
> ---
>
> **W1-1**: "linear correlation" is a very strong term and may be inappropriate here; the authors likely mean "monotonicity" instead of "linearity."
>
> **Response**: We appreciate your suggestion. We will revise the wording in the final version of the paper.
>
> ---
>
> **W1-2**: The current experiments are insufficient to draw definitive conclusions. The results are not conducted over multiple rounds, nor do they report the mean and standard deviation of performance, which makes them highly dependent on the seed, especially in the Dirichlet instance split.
>
> **Response**: We have conducted additional experiments to address your concerns. For each setting, we perform the experiments 3 times with different random seeds (42, 43, 44）and report the average performance and the standard deviation in each domain. For each domain, we apply k-means clustering to the dataset of that domain and use the pseudo-labels to construct different levels of heterogeneity.
>
> [**Table R1.** Performance (%) of different heterogeneity in each domain. The client number is 10 and 2 clients are selected in each round. We report the average performance and the standard deviation.]
> | Metric  | $\alpha$ = 10  | $\alpha$ = 1  | $\alpha$ = 0.1       | $\alpha$ = 0.01      |
> |:---------:|:--------------------:|:-------------------:|:-------------------:|:-------------------:|
> | H-Eval  | 36.17 ± 1.03      | 34.54 ± 0.35      | 32.71 ± 2.75      | 34.75 ± 2.65      |
> | M-Med   | 71.80 ± 0.84      | 71.60 ± 0.70      | 71.33 ± 0.37      | 71.56 ± 1.37      |
> | FPB     | 66.41 ± 3.31      | 68.72 ± 4.54      | 68.39 ± 2.65      | 70.07 ± 3.50      |
> | FiQA    | 22.90 ± 9.34      | 32.48 ± 9.85     | 26.42 ± 6.67      | 38.42 ± 8.78     |
> | TFNS    | 69.45 ± 2.34      | 72.99 ± 3.20      | 71.65 ± 1.64      | 73.66 ± 2.90      |
> | GSM8K   | 56.51 ± 0.14      | 59.28 ± 0.38      | 56.75 ± 0.23      | 57.64 ± 0.29      |
>
> ---
>
> **W1-3**: The number of clients (10, with 2 clients involved in each round) is relatively small and may not be significantly affected by $\alpha$.
>
> **Response**: We have conducted additional experiments by increasing the number of clients to 100. In each round, two clients are randomly selected for federated instruction tuning using FedAvg. The experimental results are presented as follows.
>
> [**Table R2.** Performance (%) of different heterogeneity in each domain. The client number is 100 and 2 clients are selected in each round. We report the average performance and the standard deviation.]
> | Metric  | $\alpha$ = 10        | $\alpha$ = 1         | $\alpha$ = 0.1       | $\alpha$ = 0.01      |
> |:---------:|:--------------------:|:-------------------:|:-------------------:|:-------------------:|
> | H-Eval  | 36.57 ± 1.84      | 36.57 ± 3.48      | 34.12 ± 0.26      | 36.88 ± 0.13      |
> | M-Med   | 70.73 ± 0.37      | 72.20 ± 0.45      | 71.83 ± 0.37      | 71.60 ± 0.45      |
> | FPB     | 67.87 ± 1.13      | 64.10 ± 2.07      | 66.71 ± 1.97      | 67.59 ± 1.42      |
> | FiQA    | 33.44 ± 3.01      | 19.87 ± 5.83      | 33.93 ± 3.54      | 33.69 ± 6.58      |
> | TFNS    | 72.22 ± 0.74      | 70.13 ± 2.59      | 73.01 ± 0.22      | 71.99 ± 0.95      |
> | GSM8K   | 54.32 ± 0.41      | 51.27 ± 0.28      | 58.94 ± 0.36      | 57.81 ± 0.19      |
>
> ---
>
> **W1-4**: Averaging performance across multiple datasets or tasks is problematic.
>
> **Response**: Indeed, directly averaging the performance across multiple datasets or tasks may not fully reflect the effectiveness of a method. However, the averaging across different tasks here is merely used as a tool to help compare the performance of different methods. Our conclusions and the effectiveness of FedDCA are not solely based on the comparison of average values, as we also present the performance of each individual task or dataset. Even when comparing each task individually, the experimental results still demonstrate the effectiveness of FedDCA.

---

> > ### Author Response · Authors · 2024-11-20
> >
> > **W2-1**: The assumption that there exists a public dataset requires further clarification regarding its prevalence in practice.
> >
> > **Response**: Please refer to the global comments for a detailed explanation of the necessity of using public datasets.
> >
> > ---
> >
> > **W2-2**: It is also necessary to determine how much of the improvement is due to the additional data versus the algorithmic design.
> >
> > **Response**: You might have overlooked Table 2, which can be divided into two parts, namely, unaugmented and augmented methods. The data augmentation strategies all sampled the same amount of public data and then perform federated instruction tuning. Therefore, I believe that the comparison of model performance trained with different data augmentation strategies in Table 2 has sufficiently demonstrated that, under the same amount of sampled public data, the effectiveness of FedDCA comes from our proposed domain coverage augmentation method rather than from more training data.
> >
> > Meanwhile, in Appendix A.9, we investigate the impact of data augmentation based on different retrieval data volumes on model performance. The model's performance and the corresponding domain coverage results are shown in Fig. 9. We conducted experiments with retrieval settings of [1k, 2k, 3k, 4k, 5k]. As illustrated in the figure, apart from the code domain, even the smallest data volume setting of 1k shows significant improvement compared to using only client-local data.
> >
> > ---
> >
> > **W3**: It is unclear which baseline uses this large, powerful public dataset for fine-tuning.
> >
> > **Response**: Following your suggestion, we have conducted additional experiments based on the mentioned two settings:
> >
> > **Setting A**: Using public dataset to perform fine-tuning for 3 epochs.
> >
> > **Setting B**: Further federated fine-tuning is performed using client-local data. Consistent with the construction method of local datasets in the paper, the local data for each client was randomly sampled from the training set corresponding to its domain.
> >
> > [**Table R3.** Performance of setting A and setting B in each domain.]
> > | **Setting** | **H-Eval** | **M-Med** | **FPB**  | **FiQA** | **TFNS** | **GSM8K** |
> > |:-----------:|:----------:|:---------:|:--------:|:--------:|:--------:|:---------:|
> > | **A** | 0.3841     | 0.691     | 0.7533   | 0.4109   | 0.7160   | 0.5375    |
> > | **B** | 0.4390     | 0.643     | 0.7797   | 0.6654   | 0.7826   | 0.5640    |
> >
> > ---
> >
> > **W4**: The experimental setting is unrealistic.
> >
> > **Response**: Please refer to the global comments for a detailed explanation.

---

> > > ### Author Response · Authors · 2024-11-20
> > >
> > > **M1**: The format of Table 1 is difficult to read.
> > >
> > > **Response**: The column names in Table 1 represent the methods, while the row names represent the metrics corresponding to different domains. Specifically, HumanEval is a commonly used benchmark for the Code domain, and MMLU-Med is frequently used to evaluate model performance in the Medical domain. FPB, FiQA, and TFNS are utilized to test the model's performance in the Financial domain, and GSM8K is a widely adopted benchmark for the Mathematical domain. For more detailed information about the datasets and metrics, please refer to Section 5.1.
> > >
> > > ---
> > >
> > > **Q1**: Does random sampling baseline make a sample of each client's domain data?
> > >
> > > **Response**: In Table 1, "Random Sampling" does not refer to sampling from the client's local data. Instead, it refers to the server randomly sampling a portion of the public data for each client to perform data augmentation.
> > >
> > > ---
> > >
> > > **Q2**: In "Results demonstrate that domain coverage significantly influences model performance in the corresponding domain," is this conclusion drawn from the experiments in this manuscript or from Wang et al. (2023)?
> > >
> > > **Response**: I am not sure which article by Wang you are referring to. In our paper, we mentioned that our exploration of the impact of domain coverage on FedDIT was inspired by Explore-Instruct [1]. However, drawing similar conclusions does not mean that we directly borrow the findings of Explore-Instruct. Whether these conclusions hold true in the context of FedDIT remains an unexplored question, requiring extensive experimentation for validation. This is precisely what this paper addresses. Based on experimental results, we demonstrate that under FedDIT, domain coverage significantly influences the performance of models in the corresponding domain.
> > >
> > > Specifically, our understanding of domain coverage differs from Explore-Instruct in the following ways:
> > >
> > > 1) **Different Definitions of Coverage**
> > >    - In Explore-Instruct, domain coverage is reflected through the distribution of verb-noun pairs, which represents the dataset's coverage of a specific domain.
> > >    - In contrast, our paper defines domain coverage using the facility location function [2], as detailed in Eq. 5. We compute domain coverage based on cosine similarity between embeddings as a distance metric. This method is more general than verb-noun pairs. For example, in the code domain, the verb-noun pairs metric is not applicable, whereas our embedding-based method can still effectively calculate domain coverage.
> > >
> > > 2) **Different Approaches to Coverage Enhancement**
> > >    - Explore-Instruct employs a DFS (Depth-First Search) approach, where forward exploration helps expand the task space by exploring more potential instructions and task variants. Backtracking allows the model to avoid incorrect exploration paths by returning and adjusting strategies, thereby avoiding local optima or ineffective paths. This combined approach enables DFS to effectively enhance instruction coverage for specific domains while maintaining flexibility and accuracy, ensuring valuable tasks or instructions are not missed.
> > >    - In contrast, FedDCA employs a **greedy client-center selection strategy** to maximize the coverage of augmented data in the latent space over in-domain data.
> > >
> > > By implementing these differences, we provide a more generalized approach to domain coverage, particularly suited for federated learning settings. The methodology and findings of this paper address specific gaps that Explore-Instruct did not cover, further validating the importance of domain coverage in federated learning scenarios.
> > >
> > > [1] Wan, Fanqi, et al. "Explore-instruct: Enhancing domain-specific instruction coverage through active exploration." arXiv preprint arXiv:2310.09168 (2023).
> > >
> > > [2] Cornuéjols, Gérard, George Nemhauser, and Laurence Wolsey. The uncapicitated facility location problem. Cornell University Operations Research and Industrial Engineering, 1983.
> > >
> > > ---
> > >
> > > Overall, we hope that our responses can fully address your concerns and will be grateful for any feedback.

---

> ### Comment · Reviewer_8vFk · 2024-11-20
> **Response to Rebuttal**
>
> I appreciate the authors' detailed response and additional experiments. However, many concerns remain unresolved.
>
> 1. Table R3 seems to support my concerns regarding the large public dataset. The rebuttal lacks comparison or analysis, so I manually compared Table R3 with Table 2. I found that FedDCA performs worse than both Baselines A and B on FPB, FiQA, and GSM8K, and worse than Baseline B on H-Eval and TFNS. This suggests that having access to a large public dataset can easily lead to superior performance. Additionally, it is unclear whether Model A has converged, given that only three epochs were trained, implying that further improvements on the baseline are possible.
>
> 2. The explanation regarding the public dataset remains unclear. The authors should either provide specific real-world cases where a large public dataset is available or demonstrate that FedDCA performs well under distribution shifts. For the first point, no specific practical application was introduced. Regarding the second point, while it is a good start, the experimental setting is not clearly described - the details of the public dataset, such as its size and distribution compared to client datasets, are not provided.
>
> 3. In response to W2-2, I am aware of Table 2 and understand that FedDCA is more effective with the same amount of sampled public data. However, the question is why sampling from the public data is necessary. The results in Table R3 indicate that direct fine-tuning with the full dataset yields superior performance compared to a complex federated learning approach.

---

> > ### Author Response · Authors · 2024-11-21
> >
> > Thank you for your time and valuable suggestions. Here are our detailed responses.
> >
> > ---
> >
> > **C1**: Table R3 seems to support my concerns regarding the large public dataset.
> >
> > **Response**: I refrained from making extensive comments because I am not entirely clear about the motivation behind your request to conduct these two experiments, so I want first to see your response. Table R3, while supporting your concern, also aligns with the scaling law [1]. If FedDCA outperforms both Setting A and Setting B across the board, then perhaps I should first reconsider the validity of the scaling law.
> >
> > First, as explained in the global comment about public datasets, public datasets are abstractions of open-source data from the internet. Hence, your request to fine-tune on public data at the server essentially translates to fine-tuning on all internet data. Have you considered the time and computational costs involved in this, including the training time that clients can tolerate? Without addressing these aspects, discussing federated learning becomes meaningless.
> >
> > I disagree with your proposal to use Setting A and Setting B as baselines, as this is not a fair comparison. Even with each client sampling 5,000 samples, it only accounts for about 12.5% of the public dataset, and that’s without considering the potential overlap in augmented datasets among clients. Instead of engaging in a data volume "arms race", I believe it would be more convincing if you could propose other, more effective data augmentation methods.
> >
> > Additionally, rather than simply adhering to the scaling law, we reiterate the necessity of effective data augmentation is to prevent potential performance degradation caused by relying solely on local data and to enhance the model’s generalization ability. As shown in Table 1 of the paper, Table R1 from the global comment’s out-of-task setting experiments, and Table R3 from the Setting A and B experiments, training with only local data or inappropriate data augmentation inevitably faces potential performance degradation, which can not be denied.
> >
> > To further illustrate the effectiveness of FedDCA and have a fair competition, we added Setting C and Setting D. These involve evaluating the performance when each client randomly samples 100 data or samples 100 data using FedDCA and performs federated instruction tuning, based on Setting A.
> >
> > [**Table C1.** Performance of setting A, B, C and D in each domain.]
> > | Setting   | H-Eval  | M-Med   | FPB     | FiQA    | TFNS    | GSM8K   |
> > |:---------:|:-------:|:-------:|:-------:|:-------:|:-------:|:-------:|
> > | Setting A | 0.3841  | 0.691   | 0.7533  | 0.4109  | 0.7160  | 0.5375  |
> > | Setting B | 0.4390  | 0.643   | 0.7797  | 0.6654  | 0.7826  | 0.5640  |
> > | Setting C | 0.4146  | 0.669   | 0.7450  | 0.2981  | 0.7051  | 0.5701  |
> > | Setting D | 0.4412  | 0.709   | 0.8143  | 0.7200  | 0.7881  | 0.5890  |
> >
> > As shown in Table C1, even with only 100 samples selected via FedDCA, the method not only prevents performance degradation but also helps the model achieve better generalization within the domain. This is attributed to exposure to more diverse in-domain data during training. Meanwhile, it can be observed that random sampling still leads to performance degradation compared to Setting A, and even performs worse than Setting B, where no data augmentation was applied. In the financial domain, performance degradation is also evident.
> >
> > In conclusion, this experiment further highlights the necessity and effectiveness of FedDCA and designing efficient data augmentation algorithms in FedDIT.
> >
> > [1] Kaplan, Jared, et al. "Scaling laws for neural language models." arXiv preprint arXiv:2001.08361 (2020).
> >
> > ---
> >
> > **C2**: The explanation regarding the public dataset remains unclear.
> >
> > **Response**: I hope you can take some time to review the global comments, which include explanations regarding the public dataset setting. Additionally, we have conducted experiments addressing the second point you mentioned about distribution shifts. As shown in Table R1 of the global comment, even when the public data is out-of-task, FedDCA still achieves performance improvements compared to other baselines.
> >
> > ---
> >
> > **C3**: In response to W2-2, I am aware of Table 2 and understand that FedDCA is more effective with the same amount of sampled public data. However, the question is why sampling from the public data is necessary.
> >
> > **Response**: Please refer to the responses provided for C1 and C2 above.
> >
> > ---
> >
> > Overall, we hope that our responses can fully address your concerns and will be grateful for any feedback.

---

> ### Comment · Reviewer_8vFk · 2024-11-21
> **Further Response to Authors' Response**
>
> The performance of fine-tuning with all available public data suggests the necessity of federated learning. In practice, federated learning is considered usually when: (1) the amount of public data is insufficient, or (2) the public data exhibits a different distribution from that of the clients. If fine-tuning using public data already yields superior performance, it implies that public data is more valuable than clients' data, thereby **eliminating the need for federated learning**. Although full-data fine-tuning may not be an efficient baseline, it is mentioned in the review due to its straightforward implementation and ease of evaluation.
>
> When clients' data is less significant, there are numerous **centralized** data-efficient [1,2,3,4,5] fine-tuning methods to enhance efficiency of fine-tuning public data. A simple approach is to cluster the data and use only the cluster centers for fine-tuning. These data-efficient fine-tuning methods can achieve comparable or even better performance than full-data fine-tuning and significantly outperform the *random sampling* baseline used in the manuscript.
>
> **References**
>
> [1] Lin, Xinyu, et al. "Data-efficient Fine-tuning for LLM-based Recommendation." Proceedings of the 47th International ACM SIGIR Conference on Research and Development in Information Retrieval. 2024.
>
> [2] Chen, Hao, et al. "Maybe only 0.5% data is needed: A preliminary exploration of low training data instruction tuning." arXiv preprint arXiv:2305.09246 (2023).
>
> [3] Das, Devleena, and Vivek Khetan. "DEFT: Data Efficient Fine-Tuning for Large Language Models via Unsupervised Core-Set Selection." arXiv preprint arXiv:2310.16776 (2023).
>
> [4] Pan, Jing, et al. "Cosmic: Data efficient instruction-tuning for speech in-context learning." arXiv preprint arXiv:2311.02248 (2023).
>
> [5] Liu, Zikang, et al. "Less is More: Data Value Estimation for Visual Instruction Tuning." arXiv preprint arXiv:2403.09559 (2024).

---

> > ### Author Response · Authors · 2024-11-21
> >
> > **Response**: Thank you for your detailed response. However, we respectfully disagree with certain points raised.
> >
> > First, we noticed that the response did not directly address our original comment but instead focused on explaining the motivation for proposing Settings A and B. We believe it is essential to engage directly with the core concerns we raised for clarity and mutual understanding.
> >
> > Second, we want to clarify that we have never suggested that client data is unimportant. Regarding scenarios where public data and client data distributions are inconsistent (e.g., out-of-task cases), we emphasize that client data remains critical. The primary necessity for data augmentation lies in preventing potential performance degradation caused by relying solely on local data and enhancing the model's generalization capabilities. This necessity is not merely a matter of following scaling laws but a practical strategy for achieving robust performance.
> >
> > We appreciate your acknowledgment of related works on data selection for efficient fine-tuning, as we explicitly cited one of the representative methods in our paper. For the data augmentation baselines, in addition to random sampling and dense retrieval, we also explored two distinct approaches:
> >
> > - Gradient-based similarity retrieval: LESS [1].
> > - Generative methods: Self-Instruct [4] and FewFedPIT [3].
> >
> > LESS, originally designed for selecting influential data for efficient instruction fine-tuning, was employed in Table 2 as a data augmentation strategy. From the client's perspective, this represents data augmentation, while from the server's perspective, it can be viewed as data selection. We believe that data selection and data augmentation [2-8] are not contradictory but complementary: one focuses on improving data quality, while the other enhances diversity. This synergy can significantly boost overall performance. Thus, it is not about choosing one over the other but leveraging both effectively, as demonstrated by Setting D in Table C1.
> >
> > Finally, we want to reiterate that **FedDCA is a plug-and-play framework** that does not preclude other methods. Its design allows seamless integration with different strategies, highlighting its flexibility and scalability.
> >
> > [1] Xia, Mengzhou, et al. "LESS: Selecting Influential Data for Targeted Instruction Tuning." Forty-first International Conference on Machine Learning.
> >
> > [2] Wang, WenHao, et al. "KnowledgeSG: Privacy-Preserving Synthetic Text Generation with Knowledge Distillation from Server." Proceedings of the 2024 Conference on Empirical Methods in Natural Language Processing. 2024.
> >
> > [3] Zhang, Zhuo, et al. "Fedpit: Towards privacy-preserving and few-shot federated instruction tuning." arXiv preprint arXiv:2403.06131 (2024).
> >
> > [4] Wang, Yizhong, et al. "Self-Instruct: Aligning Language Models with Self-Generated Instructions." Proceedings of the 61st Annual Meeting of the Association for Computational Linguistics (Volume 1: Long Papers). 2023.
> >
> > [5] Taori, Rohan, et al. Stanford Alpaca: An Instruction-Following LLaMA Model. 2023. GitHub, https://github.com/tatsu-lab/stanford_alpaca.
> >
> > [6] Wan, Fanqi, et al. "Explore-Instruct: Enhancing Domain-Specific Instruction Coverage through Active Exploration." Proceedings of the 2023 Conference on Empirical Methods in Natural Language Processing. 2023.
> >
> > [7] He, Hongliang, et al. "OpenWebVoyager: Building Multimodal Web Agents via Iterative Real-World Exploration, Feedback and Optimization." arXiv preprint arXiv:2410.19609 (2024).
> >
> > [8] Yuan, Weizhe, et al. "Self-Rewarding Language Models." Forty-first International Conference on Machine Learning.
> >
> > ---
> >
> > Overall, we hope that our responses can fully address your concerns and will be grateful for any feedback.

---

### Official Review · Reviewer_tYyT · 2024-10-31

**Soundness:** 2
**Presentation:** 3
**Contribution:** 2
**Rating:** 3
**Confidence:** 4

**Summary:**

This work proposes a federated domain-specific instruction tuning method to enable federated clients to utilize data from the server to augment their local data. The objective of this work is meaningful, as FL does indeed face the issue of insufficient data on the client-side. However, the assumption that there exists a large dataset on the server encompassing all domains seems unrealistic. If such a dataset is available on the server, why would there still be a need for FL to tune LLMs? Additionally, compared to traditional FL, this approach poses a risk of exposing additional information from the client-side data (such as domain information) to the server. In summary, I believe this work requires further refinement.

**Strengths:**

1. This manuscript is well-written and well-organized, making this work easy to understand.
2. Extensive experiment evaluations are provided.

**Weaknesses:**

1. The main distinction between the proposed method and previous approaches is its ability to leverage server-side data, with the effectiveness of the method relying on the server holding a dataset that encompasses multiple domains. In fact, many LLM tuning methods based on FL aim to explore the possibility of tuning LLMs with client-side data when centrally collected data has been exhausted. If there is already a dataset on the server that encompasses multiple domains, why would we still need FL to tune the model? Wouldn't it be better to fine-tune the LLM directly on the central server?
2. In the experiments, it is mentioned that "each client has 100 local instructions and obtains 5000 augmented public instructions from the server." (line 885). In this case, the majority of the data is contributed by the server, and the significance of introducing FL seems unclear.
3. This work builds upon the original FL paradigm by providing additional client-side information, specifically client-side cluster centers. Although the original data remains on the client side, it is inappropriate to directly claim that "it does not violate the client’s privacy." Such a kind of information may reveal the distribution of client-side data, thereby exposing client data to privacy leakage threats such as membership inference attacks, which do not require direct access to the original data. As FedDCA describes as a distinguishing feature from other existing methods, it sends data from the server's public dataset that are related to a specific client to that client as augmentation. Doesn’t this imply that the server can determine the data domains of that client to some extent?
4. One of the claimed contribution, "no linear correlation between the degree of heterogeneity and model performance" has already been revealed by one of the cited works [1], which weakens the contribution of this work to a certain extent.
5. To the best of my knowledge, the appropriate citation of FedIT should be [2], which has been proposed in 2023, rather than (Ye et al., 2024b;a) in Line 764.
6. The codes are not available, making the reproducibility of this work difficult to assess.

[1] FederatedScope-LLM: A Comprehensive Package for Fine-tuning Large Language Models in Federated Learning

[2] Towards Building the Federated GPT: Federated Instruction Tuning.

**Questions:**

1. How are the four baselines based on FedIT implemented? Do they involve directly tuning and transmitting all parameters, or only PEFT adapters?
2. How is the weight of the projector in FedDIT* obtained? It seems that many details have not been provided in Appendix A.4.
3. Please refer to Weaknesses.

---

> ### Author Response · Authors · 2024-11-20
>
> Thank you for your time and valuable suggestions. Here are our detailed responses.
>
> ---
>
> **W1**: Questions about the setting of using public datasets for data augmentation.
>
> **Response**: For the necessity of using public datasets, please refer to the global comments for a detailed explanation.
>
> ---
>
> **W2**: The significance of introducing FL seems unclear.
>
> **Response**: You might have overlooked Table 2, which can be divided into two parts, namely, unaugmented and augmented methods. The data augmentation strategies all sampled the same amount of public data and then perform federated instruction tuning. Therefore, I believe that the comparison of model performance trained with different data augmentation strategies in Table 2 has sufficiently demonstrated that, under the same amount of sampled public data, the effectiveness of FedDCA comes from our proposed domain coverage augmentation method rather than from more training data.
>
> Meanwhile, in Appendix A.9, we investigate the impact of data augmentation based on different retrieval data volumes on model performance. The model's performance and the corresponding domain coverage results are shown in Fig. 9. We conducted experiments with retrieval settings of [1k, 2k, 3k, 4k, 5k]. As illustrated in the figure, apart from the code domain, even the smallest data volume setting of 1k shows significant improvement compared to using only client-local data.
>
> ---
>
> **W3**: Privacy concerns.
>
> **Response**: Firstly, transmitting cluster centers [1,2] may indeed enable the server to infer the data domain of a client. Secondly, your concern essentially revolves around whether public datasets should be used for data augmentation. This is because, with public dataset based augmentation, the server could deduce the client's data domain through the data retrieved by the client, except in strategies like random sampling.
>
> There are also related works that avoid using public datasets for data augmentation, such as FewFedPIT [3], which performs data augmentation via self-Instruct locally on the client. We include an enhanced version of this method (performs self-Instruct with GPT) as a baseline in our experiments, as shown in Table 2. The results indicate a significant performance gap between this baseline and FedDCA. We attribute this gap to the fact that pre-trained LLMs cannot provide effective, stable, and high-quality instruction generation. Moreover, generating instructions locally is very costly in terms of both time and computational resources.
>
> Therefore, I believe there is a tradeoff between leveraging public datasets (with the risk of potential data domain leakage) and achieving better model performance.
>
> [1] Zhang, Jianqing, et al. "Fedtgp: Trainable global prototypes with adaptive-margin-enhanced contrastive learning for data and model heterogeneity in federated learning." Proceedings of the AAAI Conference on Artificial Intelligence. Vol. 38. No. 15. 2024.
>
> [2] Tan, Yue, et al. "Fedproto: Federated prototype learning across heterogeneous clients." Proceedings of the AAAI Conference on Artificial Intelligence. Vol. 36. No. 8. 2022.
>
> [3] Zhang, Zhuo, et al. "Fedpit: Towards privacy-preserving and few-shot federated instruction tuning." arXiv preprint arXiv:2403.06131 (2024).

---

> > ### Author Response · Authors · 2024-11-20
> >
> > **W4**: "No linear correlation between the degree of heterogeneity and model performance" has already been revealed by one of the cited works [1].
> >
> > **Response**: This point may be misleading. FederatedScope [1] compares the impact of different heterogeneity levels on personalized federated learning LLMs in Section 6.4, concluding that as the degree of heterogeneity decreases, the performance of federated fine-tuning based on FedAvg gradually approaches that of the global scenario. This directly contradicts the reviewer's claim that "there is no linear correlation between the degree of heterogeneity and model performance,". To elaborate further, there are significant differences and limitations between their study and ours:
> >
> > **1) Limited Domain.** FederatedScope only studies the code domain and does not further explore other domains, which weakens the persuasiveness of its conclusions to some extent. In contrast, FedDCA conducts extensive experiments across four domains (code, medical, financial, and mathematical) as well as multi-domain settings.
> >
> > **2) Ungeneralized Heterogeneity Setting.** FederatedScope constructs heterogeneity using a Dirichlet distribution based on programming languages as labels, which is only effective for the code domain. It does not consider effective ways to construct heterogeneous distributions for other domains. In contrast, FedDCA adopts a domain-agnostic labeling approach, where it maps samples to a latent space via an encoder, performs clustering, and uses the clusters as pseudo-labels. We believe this pseudo-labeling approach provides a more general and flexible way to construct heterogeneous data distributions.
> >
> > **3) Different Conclusions.** FederatedScope concludes that as the degree of heterogeneity decreases, the performance of federated fine-tuning based on FedAvg gradually approaches that of the global scenario. This conclusion is fundamentally different from that of FedDCA, which demonstrates through extensive experiments across different settings and domains that there is no linear correlation between the degree of heterogeneity and model performance.
> >
> > In conclusion, we do not believe that our experiments on data heterogeneity are related to the experiments conducted in FederatedScope. The differences are significant across the choice of domains, the methods for constructing heterogeneous data distributions, and the final conclusions.
> >
> > [1] Kuang, Weirui, et al. "Federatedscope-llm: A comprehensive package for fine-tuning large language models in federated learning." Proceedings of the 30th ACM SIGKDD Conference on Knowledge Discovery and Data Mining. 2024.
> >
> > ---
> >
> > **W5**: The appropriate citation of FedIT should be Shepherd [2].
> >
> > **Response**: We do cite Shepherd [2] in the introduction. FedDCA is mainly implemented based on the OpenFedLLM framework [1], which is why it primarily references OpenFedLLM. A more appropriate approach would be to cite OpenFedLLM and Shepherd simultaneously.
> >
> > [1] Ye, Rui, et al. "Openfedllm: Training large language models on decentralized private data via federated learning." Proceedings of the 30th ACM SIGKDD Conference on Knowledge Discovery and Data Mining. 2024.
> >
> > [2] Zhang, Jianyi, et al. "Towards building the federatedGPT: Federated instruction tuning." ICASSP 2024-2024 IEEE International Conference on Acoustics, Speech and Signal Processing (ICASSP). IEEE, 2024.
> >
> > ---
> >
> > **W6**: The codes are not available.
> >
> > **Response**: We have released the implementation of FedDCA as the supplementary material now.

---

> > > ### Author Response · Authors · 2024-11-20
> > >
> > > **Q1-1**: How are the four baselines based on FedIT implemented?
> > >
> > > **Response**: Our experiments are implemented using the OpenFedLLM framework, which adheres to the original implementation approaches for FedAvg [1], FedProx [2], SCAFFOLD [3], and FedAvgM [4]. The detailed implementation can be found in the OpenFedLLM repository at https://github.com/rui-ye/OpenFedLLM.git, as well as in the corresponding papers for these four methods.
> > >
> > > [1] McMahan, Brendan, et al. "Communication-efficient learning of deep networks from decentralized data." Artificial intelligence and statistics. PMLR, 2017.
> > >
> > > [2] Li, Tian, et al. "Federated optimization in heterogeneous networks." Proceedings of Machine learning and systems 2 (2020): 429-450.
> > >
> > > [3] Karimireddy, Sai Praneeth, et al. "Scaffold: Stochastic controlled averaging for federated learning." International conference on machine learning. PMLR, 2020.
> > >
> > > [4] Hsu, Tzu-Ming Harry, Hang Qi, and Matthew Brown. "Measuring the effects of non-identical data distribution for federated visual classification." arXiv preprint arXiv:1909.06335 (2019).
> > >
> > > ---
> > >
> > > **Q1-2**: Do they involve directly tuning and transmitting all parameters or only PEFT adapters?
> > >
> > > **Response**: All experiments in this paper are based on LoRA for instruction tuning. The content transmitted between the client and the server during training is shown in the third figure of Figure 1. Since the parameters of the pre-trained LLM are frozen, only the adapter parameters are transmitted during training.
> > >
> > > ---
> > >
> > > **Q2**: How is the weight of the projector in FedDIT* obtained?
> > >
> > > **Response**: In Section 4.3, we describe the training method for the projector. As shown in Eq. 4, we utilize contrastive learning on the public dataset to map the output of the client's small encoder into the same latent space as the server's large encoder. For more details, please refer to the corresponding code we released.
> > >
> > > ---
> > >
> > > Overall, we hope that our responses can fully address your concerns and will be grateful for any feedback.

---

> > > > ### Comment · Reviewer_tYyT · 2024-11-20
> > > >
> > > > Thank you for your response. I still need more time to further review the other parts. As for your replies to W2 and W3, it seems my concerns have not been fully addressed.
> > > >
> > > > 1. W2: Actually, I meant that the meaning of introducing of FL for LLM tuning seems to be unclear, since this work assumes that the server holds a huge amount of data, whose domains coverage those of client-side data. From your experimental setup, the number of clients to 10 and each client has 100 local instructions, while the server holds 5,000 data samples. The amount of data held on the server is significantly greater than the total amount of data held across all clients. Given the abundance of data on the server, why is FL still necessary for fine-tuning? Have you considered directly fine-tuning using the data on the server? As shown in Figure 9, performance decreases as the amount of retrieved data is reduced. This further indicates that the performance gain of the proposed method over the ones without augmentation originates from the large volume of data on the server.
> > > > 2. W3: As you acknowledge, there is a tradeoff between leveraging public datasets and achieving better model performance. If the improvement in accuracy achieved by your method comes at the cost of compromising privacy protection, FL is not the only privacy-preserving machine learning method available. Transforming client-side data into forms that minimize retained private information can also serve the purpose of enhancing LLMs using client-side data. Without further comparisons with other privacy-preserving approaches, it is difficult to assess the significance of your method.

---

> > > > > ### Author Response · Authors · 2024-11-21
> > > > >
> > > > > **C1-1**: The amount of data held on the server is significantly greater than the total amount of data held across all clients.
> > > > >
> > > > > **Response**: We conduct the following experiment: each client sampled 100 samples from the public dataset and then performs FedDIT. The performance across domains is shown in Table C1. Here, Random$^*$ and FedDCA$^*$ represent the default setting, where each client samples 5,000 samples.
> > > > >
> > > > > [**Table C1.** Performance (%) of FedDCA and Random Sampling with different amounts of sampled public data. Random is short for Random Sampling.]
> > > > > | Method  | H-Eval | MMLU-Med | FPB   | FiQA  | TFNS  | GSM8K |
> > > > > |:--------:|:------:|:--------:|:-----:|:-----:|:-----:|:-----:|
> > > > > | Base Data| 39.03  | 68.40    | 58.25 | 14.18 | 66.62 | 47.46 |
> > > > > | Random   | 34.53  | 69.80    | 60.89 | 14.54 | 65.57 | 48.77 |
> > > > > | Random*  | 32.93  | 71.30    | 64.19 | 13.09 | 65.53 | 47.38 |
> > > > > | FedDCA   | 35.97  | 70.20    | 63.20 | 15.63 | 67.58 | 49.32 |
> > > > > | FedDCA*  | 36.58  | 74.50    | 67.24 | 35.27 | 73.32 | 52.46 |
> > > > >
> > > > > The results show that even with a small amount of sampled data, the model's performance still improves compared to Random sampling of the same data amount, except in the code domain, which tends to follow a certain paradigm, as mentioned in the paper. This further demonstrates the effectiveness of FedDCA. Since all our comparisons are based on sampling the same amount of public data, the effectiveness of FedDCA is independent of the sampling scale.
> > > > >
> > > > > Also, my response to reviewer 8vFk (Table C1) may provide additional evidence to support the necessity of sampling from the public dataset.
> > > > >
> > > > > ---
> > > > >
> > > > > **C1-2**: Given the abundance of data on the server, why is FL still necessary for fine-tuning? Have you considered directly fine-tuning using the data on the server?
> > > > >
> > > > > **Response**:  We re-emphasize the necessity of data augmentation is to prevent potential performance degradation caused by relying solely on local data and to enhance the model's generalization ability. My response to reviewer 8vFk (Table C1) may provide additional evidence to support the necessity of sampling from the public dataset.
> > > > >
> > > > > ---
> > > > >
> > > > > **C2**: As you acknowledge, there is a tradeoff between leveraging public datasets and achieving better model performance.
> > > > >
> > > > > **Response**: What I acknowledge is that, in the case where public data is on the server, there exists a tradeoff between privacy and performance. However, as I explained in the global comment about public data, public data is an abstraction of internet data. The core innovation of this work lies in maximizing domain coverage, and the proposed algorithm is independent of the presence of a public dataset on the server. Even if the server does not have a public dataset, clients can upload their cluster centers to the server, which selects a set of client centers and sends them back to the clients. Each client can then retrieve data from the website based on the received client centers by itself, thereby achieving data augmentation that maximizes domain coverage. In this case, since the server does not know the encoder used by the client, it can not infer the semantic meaning of the embedding itself from the features. For the server, it becomes significantly more challenging to infer the client’s domain, let alone apply any privacy protection techniques to the embeddings.
> > > > >
> > > > > To validate my idea, I have provided two examples for illustration. We used two different encoders: `'BAAI/bge-large-en-v1.5'` (denoted as $w_1$) and `'google-bert/bert-large-uncased'` (denoted as $w_2$). Both encoders output 1024-dimensional features.
> > > > >
> > > > > **Example 1**: Both $w_1$ and $w_2$ take "hello world" as input, and the cosine similarity between their embeddings is **0.1829**.
> > > > >
> > > > > **Example 2**: Three instructions were tested:
> > > > > - **Instruction 1**: "Create an array of length 5 which contains all even numbers between 1 and 10."
> > > > > - **Instruction 2**: "Write a replace method for a string class which replaces the given string with a given set of characters."
> > > > > - **Instruction 3**: "What is the sentiment of this news? Please choose an answer from {negative/neutral/positive}. Teollisuuden Voima Oyj, the Finnish utility known as TVO, said it shortlisted Mitsubishi Heavy's EU-APWR model along with reactors from Areva, Toshiba Corp., GE Hitachi Nuclear Energy, and Korea Hydro & Nuclear Power Co."
> > > > >
> > > > > For these instructions:
> > > > > - Instruction 1 is passed to $w_1$.
> > > > > - Instructions 2 and 3 are passed to $w_2$.
> > > > >
> > > > > This results in three embeddings: $e_1$, $e_2$, and $e_3$. The cosine similarity between $e_1$ and $e_2$ is **0.1464**, while the similarity between $e_1$ and $e_3$ is **0.1879**. Instructions 1 and 2 are in the same domain, whereas have a lower cosine similarity.
> > > > >
> > > > > In conclusion, as demonstrated above, concerns regarding the server potentially inferring clients' domains can be effectively addressed.
> > > > >
> > > > > ---
> > > > >
> > > > > Overall, we hope that our responses can fully address your concerns and will be grateful for any feedback.

---

### Official Review · Reviewer_29fA · 2024-11-03

**Soundness:** 3
**Presentation:** 3
**Contribution:** 3
**Rating:** 5
**Confidence:** 4

**Summary:**

The paper proposes a new method for federate instruction fine tuning for LLMs.
The main idea involves around using cluster center info coming from clients to select publicly available data stored on the server to augment the local data to achieve better coverage.

**Strengths:**

+ The paper is motivated well.

+ The paper is easy to read.

+ Provides a basic clustering method that could improve the federated fine tuning process.

**Weaknesses:**

- To me, the main question I have is that whether FedAvg and variants developed for dealing with Non-IID data combined with simple random sampling of the public data could achieve similar results.   The main reason is that Federated learning by itself is not sending data to the clients and just aggregates the client updates. Therefore, it is not surprising to me that sending more data to clients help even if this is done carefully using clustering as suggested in the paper. Therefore, how about sending some random data to clients and then use standard FL techniques especially the ones developed for non-iid settings.    Hence, I highly recommend the authors to explore this setting. One initial experiment could be to send the random data (e.g., the same amount sent using the clustering approach) to each client and then use FedAvg to see the performance.

- The privacy experiments are good but not itself it is not adequate ( I understand that this is not the focus of this paper and I did not use this comment in any negative way in my ranking), I think a setting where noise is added during aggregation (e.g., something of differential privacy sense) can be evaluated. As a starting point, Gaussian noise could be added to achieve \epsilon-\delta differential privacy. Again this could be considered in the future work.

**Questions:**

Please see my first comment in the weaknesses section.

---

> ### Author Response · Authors · 2024-11-20
>
> Thank you for your time and valuable suggestions. Here are our detailed responses.
>
> ---
>
> **W1**: Whether FedAvg and variants developed for dealing with Non-IID data combined with simple random sampling of the public data could achieve similar results.
>
> **Response**: I'm not quite sure if I understand your point. If you are referring to using a random sampling strategy for data augmentation, Table 2 already includes this baseline, and its performance significantly lags behind that of FedDCA and FedDCA$^*$. If you are questioning the setting of using public datasets for data augmentation, this might be due to a misunderstanding arising from our relatively simple construction method of both private and public data.
>
> For the necessity of using public datasets, please refer to the global comments for a detailed explanation.
>
> **W2**: Differential privacy experiments.
>
> **Response**: We appreciate your suggestion and will consider adding differential privacy experiments in our future work.
>
> ---
>
> Overall, we hope that our responses can fully address your concerns and will be grateful for any feedback.

---

> > ### Comment · Reviewer_29fA · 2024-11-21
> >
> > Thanks for your response. There are other federated learning mechanisms dealing with non-id data that seems to perform better then FedCA.  I suggest running other federated learning algorithms in addition to using random sampling.
> >
> > Also, I read the general response on the public dataset existence. The new experiments seem to be done in a similar domain.

---

> > > ### Author Response · Authors · 2024-11-22
> > >
> > > Thank you for your time and valuable suggestions. Here are our detailed responses.
> > >
> > > ---
> > > **C1-1**: There are other federated learning mechanisms dealing with non-id data that seems to perform better then FedCA.
> > >
> > > **Response**: It seems there may be some misinterpretations in your understanding of the article. This article primarily focuses on achieving effective domain-specific instruction augmentation under the FedDIT setting, rather than attempting to solve the non-IID problem. As shown in Appendix A.2, our experiments demonstrate that cross-client domain coverage, rather than data heterogeneity, is the key factor driving model performance in FedDIT. To address this, we propose FedDCA, which optimizes the cross-client domain coverage through greedy client-center selection and retrieval-based data augmentation.
> > >
> > > In addition to random sampling and dense retrieval, we explored two additional approaches:
> > >
> > > - **Gradient-based similarity retrieval:** LESS [1].
> > > - **Generative methods:** Self-Instruct [2] and FewFedPIT [3].
> > >
> > > As shown in Table 2, **FedDCA outperforms other baselines across all domains**. Moreover, in Section 5.2, we discuss the efficiency of different methods. Other approaches either incur higher costs (e.g., Self-Instruct relies on API calls), demand greater computational overhead (e.g., FewFedPIT and LESS), or require more information (e.g., LESS). **In contrast, FedDCA achieves superior performance while maintaining better efficiency**, further underscoring its effectiveness and practicality.
> > >
> > > [1] Xia, Mengzhou, et al. "LESS: Selecting Influential Data for Targeted Instruction Tuning." Forty-first International Conference on Machine Learning.
> > >
> > > [2] Wang, Yizhong, et al. "Self-Instruct: Aligning Language Models with Self-Generated Instructions." Proceedings of the 61st Annual Meeting of the Association for Computational Linguistics (Volume 1: Long Papers). 2023.
> > >
> > > [3] Zhang, Zhuo, et al. "Fedpit: Towards privacy-preserving and few-shot federated instruction tuning." arXiv preprint arXiv:2403.06131 (2024).
> > >
> > > ---
> > >
> > > **C1-2**: The new experiments seem to be done in a similar domain.
> > >
> > > **Response**: As we mentioned in the global comments, firstly, the concept of "domain" is flexible and hierarchical. Currently, there is no precise definition of a domain. For example, in Explore-Instruct [1], "Brainstorming" and "Rewriting" are considered two domains, but they could also be regarded as two tasks under the general domain.
> > >
> > > Secondly, even when using the clearer domain classification adopted in this paper (e.g., code, medical, financial, math), if the client aims to solve tasks based on existing knowledge, the public dataset will inevitably contain knowledge relevant to those domains. This could come from the original corpus (which can be converted into instruction-response pairs using GPT [2-3]) or from pre-constructed instruction datasets. So the public dataset contains in-domain but out-of-task instructions is a more reasonable and intuitive scenario.
> > >
> > > As shown in Table R1, the results once again demonstrate the **two key roles of FedDCA: preventing potential performance degradation and enhancing the model's generalization ability.**
> > >
> > > [1] Wan, Fanqi, et al. "Explore-Instruct: Enhancing Domain-Specific Instruction Coverage through Active Exploration." Proceedings of the 2023 Conference on Empirical Methods in Natural Language Processing. 2023.
> > >
> > > [2] Zhou, Kun, et al. "JiuZhang3. 0: Efficiently Improving Mathematical Reasoning by Training Small Data Synthesis Models." arXiv preprint arXiv:2405.14365 (2024).
> > >
> > > [3] Yue, Xiang, et al. "Mammoth2: Scaling instructions from the web." arXiv preprint arXiv:2405.03548 (2024).
> > >
> > > ---
> > >
> > > Overall, we hope that our responses can fully address your concerns and will be grateful for any feedback.

---

### Official Review · Reviewer_4A8X · 2024-11-04

**Soundness:** 3
**Presentation:** 3
**Contribution:** 3
**Rating:** 6
**Confidence:** 3

**Summary:**

This paper introduces FedDCA, a novel approach to federated learning for domain-specific LLM instruction tuning that focusses on coverage, as opposed to heterogeneity (non-iid ness). Instead of worrying about how different each client's data is from others (data heterogeneity), the authors show that what really matters is how well the combined data from all clients covers the target domain. This insight leads to their proposed method which uses client selection and data augmentation strategies to maximize domain coverage.

The authors demonstrate their approach across four domains (code, medical, financial, and mathematical), showing significant improvements over existing methods. They also introduce a more computationally efficient variant called FedDCA* that uses different-sized encoders for clients and servers. While the paper lacks some theoretical guarantees and has limited privacy analysis, it presents a promising new direction for federated learning with LLMs by focusing on domain coverage rather than trying to manage data heterogeneity.

The authors do not provide any additional methods to guarantee privacy (no DP), though it is common in FL papers, also they mount extraction attacks to measure memorization.

**Strengths:**

1. novel idea and surprising insight, regarding coverage vs. heterogeneity.

2. Extensive and thorough experimentation

**Weaknesses:**

1. There is one main concern I have, maybe because I have not been following FL literature recently, but it seems like the baselines are all from 2020! I wonder why, is the setup not realistic? The paper cites Ye et al. 2024 (openfedllm) which is recent, but they only compare against in table 3, why not table 2? is it because the FedIT method there is the backbone to this method?


2. Minor: seems like clients are limited to 10, which is a bit low. Also I wonder how DP would come into play, but this is not necessary to study.

**Questions:**

Please respond to the weaknesses above.

Also

How do you isolate whether the improvements come from your domain coverage method versus simply having more training data? Did you control for total training data size when comparing methods?

---

> ### Author Response · Authors · 2024-11-20
>
> Thank you for your time and valuable suggestions. Here are our detailed responses.
>
> ---
> **W1**: Outdated baselines.
>
> **Response**: There may be some biases in your understanding of the article. This article focuses on how to perform effective domain-specific instruction augmentation under the FedDIT setting. Therefore, different FL baselines are the only strategies to perform federated instruction tuning after data augmentation, following OpenFedLLM [1]. To compare with related works of domain-specific instruction augmentation, we introduce the generation-based approach Self-Instruct [3] and discuss its weakened version FewFedPIT [2] in Section 5.2. In addition, we select LESS [4], a gradient-based similarity retrieval method, as the baseline.
>
> Specifically, FedIT is the abstraction of the FL strategies. In this article, we choose the four widely used FL strategies, including FedAvg [5], FedProx [6], SCAFFOLD [7], and FedAvgM [8]. The abstraction of FedIT in Table 3 is because these policies do not affect domain coverage. As FedIT are all unaugmented baselines and their domain coverage is calculated based on cross-client's local private data, without domain instruction augmentation.
>
> [1] Ye, Rui, et al. "Openfedllm: Training large language models on decentralized private data via federated learning." Proceedings of the 30th ACM SIGKDD Conference on Knowledge Discovery and Data Mining. 2024.
>
> [2] Zhang, Zhuo, et al. "Fedpit: Towards privacy-preserving and few-shot federated instruction tuning." arXiv preprint arXiv:2403.06131 (2024).
>
> [3] Wang, Yizhong, et al. "Self-instruct: Aligning language models with self-generated instructions." arXiv preprint arXiv:2212.10560 (2022).
>
> [4] Xia, Mengzhou, et al. "Less: Selecting influential data for targeted instruction tuning." arXiv preprint arXiv:2402.04333 (2024).
>
> [5] McMahan, Brendan, et al. "Communication-efficient learning of deep networks from decentralized data." Artificial intelligence and statistics. PMLR, 2017.
>
> [6] Li, Tian, et al. "Federated optimization in heterogeneous networks." Proceedings of Machine learning and systems 2 (2020): 429-450.
>
> [7] Karimireddy, Sai Praneeth, et al. "Scaffold: Stochastic controlled averaging for federated learning." International conference on machine learning. PMLR, 2020.
>
> [8] Hsu, Tzu-Ming Harry, Hang Qi, and Matthew Brown. "Measuring the effects of non-identical data distribution for federated visual classification." arXiv preprint arXiv:1909.06335 (2019).
>
> ---
>
> **W2**: Low client numbers.
>
> **Response**: We conducted the following extended experiments by increasing the number of clients to 100. In each round, two clients are randomly selected for federated instruction tuning using FedAvg. The experimental results are presented as follows.
>
> [**Table R1.** Performance (%) of FedDCA and other baselines in various domains. The client number is 100 and 2 clients are selected in each round. Random is short for Random Sampling.]
> | Method             | H-Eval | MMLU-Med | FPB    | FiQA   | TFNS   | GSM8K  |
> |:--------------------:|:--------:|:----------:|:--------:|:--------:|:--------:|:--------:|
> | Zero-shot          | 29.88  | 70.60    | 55.94  | 18.54  | 59.21  | 23.27  |
> | Base Data          | 34.14 | 72.40    | 66.74 | 33.45 | 72.78 | 49.12 |
> | Random  | 34.75 | 69.90    | 61.05 | 12.00 | 65.53 | 47.23 |
> | **FedDCA**             | **35.92** | **73.30**    | **67.16** | **34.51** | **73.26** | **50.26** |
>
> [**Table R2.** Domain coverage of FedDCA and other baselines in four domains. ]
> | Method             | Code   | Med.    | Fin.    | Math.   |
> |:--------------------:|:--------:|:---------:|:---------:|:---------:|
> | Base Data          | 0.8282 | 0.8377  | 0.9339  | 0.8709  |
> | Random  | 0.8685 | 0.8497  | 0.9408  | 0.8812  |
> | **FedDCA**             | **0.9242** | **0.9090**  | **0.9800**  | **0.9118** |
>
> As shown in Table R1, as the number of clients increases, the amount of local data on each client gradually grows. The model trained using only base data even outperforms Random Sampling in domains other than code and narrows the gap with FedDCA. However, as shown in Table R2, because FedDCA aims to maximize cross-client domain coverage, it achieves higher domain coverage and better performance. This experiment further demonstrates the effectiveness and scalability of FedDCA.

---

> > ### Author Response · Authors · 2024-11-20
> >
> > **Q1**: How do you isolate whether the improvements come from your domain coverage method versus simply having more training data? Did you control for total training data size when comparing methods?
> >
> > **Response**: You might have overlooked Table 2, which can be divided into two parts, namely, unaugmented and augmented methods. The data augmentation strategies all sampled the same amount of public data and then perform federated instruction tuning. Therefore, I believe that the comparison of model performance trained with different data augmentation strategies in Table 2 has sufficiently demonstrated that, under the same amount of sampled public data, the effectiveness of FedDCA comes from our proposed domain coverage augmentation method rather than from more training data.
> >
> > ---
> >
> > Overall, we hope that our responses can fully address your concerns and will be grateful for any feedback.

---

> > > ### Comment · Reviewer_4A8X · 2024-12-02
> > >
> > > Thanks for the experiments and the response. The response re-assures my assumptions on the strengths. I will keep my score.

---

> > > > ### Author Response · Authors · 2024-12-03
> > > >
> > > > Thank you for your positive feedback and for confirming the strengths of our work. We are glad to hear that our response has reassured your assumptions. Again, we appreciate your time and effort in reviewing our manuscript!

---

### Author Response · Authors · 2024-11-20

We have noticed that some reviewers have concerns about the setting of FedDCA and whether to use public datasets for data augmentation. Thus, we provide a unified response here, hoping to help you better understand and agree on the necessity of public datasets.

---

A public dataset is an abstraction of open-source datasets available on the internet. In the context of this paper, the presence of a public dataset on the server is only one possible scenario. The core innovation of this work lies in maximizing domain coverage, and the proposed algorithm is independent of the presence of a public dataset on the server. Even if the server does not have a public dataset, clients can upload their cluster centers to the server, which selects a set of client centers and sends them back to the clients. Each client can then retrieve data based on the received client centers, thereby achieving data augmentation that maximizes domain coverage. This approach was chosen to simplify the setting, much like the first step in solving an equation is simplification, allowing us to focus more on the algorithm itself.

I believe the reason why federated domain instruction fine-tuning requires data augmentation is to prevent potential performance degradation from using only local data and to enhance the model's generalization capability. I don't think the value of local data will be overshadowed by public data.

Regarding whether the public dataset must include in-domain data, first of all, the concept of "domain" is flexible and hierarchical. Currently, there is no precise definition of a domain. For example, in Explore-Instruct [1], "Brainstorming" and "Rewriting" are considered two domains, but they could also be regarded as two tasks under the general domain.

Secondly, even when using the clearer domain classification adopted in this paper (e.g., code, medical, financial, math), if the client aims to solve tasks based on existing knowledge, the public dataset will inevitably contain knowledge relevant to those domains. This could come from the original corpus (which can be converted into instruction-response pairs using GPT) or from pre-constructed instruction datasets. So the public dataset contains in-domain but out-of-task is a more reasonable and intuitive scenario.

Consider a scenario where clients hold data for solving task $t_1$, which belongs to domain $\mathcal{D}$. The public dataset includes domain $\mathcal{D}$, containing a set of tasks denoted as $T=\{t_2,\dots,t_N\}$, which does not include $t_1$. The motivation for the client to participate in federated learning is to gain additional benefits compared to purely local training; therefore, the test set should also consist of data from domain $\mathcal{D}$ and task $t_1$. In this scenario, if we do not use local data and merely train the LLM by randomly sampling from the public data, intuitively, its performance is inferior to using both local data and a portion of the sampled public data.

To further clarify, we have conducted additional experiments based on this out-of-task scenario. Considering this setting in the financial domain, to construct the out-of-task setting, given that the training set FinGPT and the test sets FPB, FiQA, and TFNS are all related to sentiment analysis tasks, we choose to continue using FPB, FiQA, and TFNS as the test sets. This means that the clients aim to train a model adept at performing sentiment analysis through federated learning. Meanwhile, the FinGPT's instructions in public data are replaced with data from the $\texttt{Sujet-Finance-Instruct-177k}$ dataset where `task_type="qa"`. The local data of the clients is still randomly sampled from FinGPT. This approach yields out-of-task public data. The following experimental results show the performance on the test set after 30 rounds of training, using Accuracy (%) as the metric.

[**Table R1**. Performance on the test set after 30 rounds of training. Random is short for Random Sampling.]
| Method | FPB | FiQA | TFNS |
|:------------:|:------------:|:------------:|:------------:|
| Zero Shot | 55.94 | 18.54 | 59.21 |
| Base Data | 58.25 | 14.18 | 66.62 |
| Random | 60.39 | 9.45 | 65.45 |
| **FedDCA** | **60.89** | **18.91** | **67.37** |

As shown in Table R1, it can be observed that even when the public data is out-of-task, FedDCA still achieves performance improvements compared to other baselines. Additionally, using the Random Sampling data augmentation strategy resulted in performance degradation on the FiQA dataset. This further underscores the necessity of selecting an appropriate data augmentation strategy.

[1] Wan, Fanqi, et al. "Explore-instruct: Enhancing domain-specific instruction coverage through active exploration." arXiv preprint arXiv:2310.09168 (2023).

---

We hope that our responses can fully address your concerns and will be grateful for any feedback.

---

### Author Response · Authors · 2024-11-24

Once again, we sincerely thank the reviewers for their valuable feedback and constructive suggestions. We have uploaded a revised version of our paper, with the changes highlighted in red for your convenience. The key modifications are as follows:

- In Section 1, we add a paragraph to briefly introduce FedDIT's setting and lead readers to Appendix A.2 for a more detailed explanation.
- We revised the description of the relationship between data heterogeneity and LLM performance, changing it from nonlinear to non-monotonic for greater precision and clarity.
- In Section 5, we add a table to highlight the key differences between FedDCA and other baselines.
- We modify the citation for FedIT, including both OpenFedLLM and Shepherd.
- In Appendix A.2, we further elaborate FedDIT's setting, including why this setting, the definition of domain and the distribution of public data.
- In Appendix A.3.1, we update the results of Table 5, which reports the average performance and the standard deviation in each domain.
- In Appendix A.6, we add the discussion of domain inference attack.
- In Appendix A.10, we perform the experiments on the held-out setting, where the public dataset does not contain the task that clients aim to solve.
- In Appendix A.10, we provide the experimental results when each client only samples 100 data from the server.
- In Appendix A.10, we provide the experimental results when the LLM is firstly fine-tuned on public data then performs FedDIT based on different settings.
- In Appendix A.10, we show the scalability of FedDCA.
- We adjust some of the figures and tables to make them more self-explanatory and to create a more compact layout for the article.

We deeply appreciate the reviewers' time and effort in evaluating this paper and hope that this revised version effectively addresses your concerns.

---

### Meta-Review · Area_Chair_rLbc · 2024-12-21

**Metareview:**

## Summary:
The paper presents a novel approach called Federated Domain Coverage Augmentation (FedDCA) for federated learning in the context of domain-specific instruction tuning for Large Language Models (LLMs). In the proposed approach, each client performs clustering of local instructions and sends the cluster centers to the server, while the server selects N (N is the number of clients) centers in a greedy manner to solve a maximum coverage problem. They assume that there is a public instruction tuning dataset and use dense retrieval to find the public instructions closest to the selected center for each client. The selected subset of the public instruction dataset is then sent to the client as an augmentation to its local private dataset for local training. The experiments conducted across four diverse domains (code, medical, financial, and mathematical) show improvement over existing federated instruction tuning methods.

## Strengths:
1. The paper raises that domain coverage is important to FedIT and proposes an idea to address it
1. Extensive comparison to baseline approaches.

## Weaknesses:
1. The assumption that there exists a public and diverse instruction-tuning dataset is not fully convincing to reviewers. If the data are of web-scale as the author claims in the response or can be easily accessed by each client, then the motivation of applying federated learning is questionable. If the public dataset is small, then it might be hard to guarantee its diversity and full coverage of all the domains. Moreover, whether federated learning is necessary and realistic in the paper-proposed setting is not well explained. Although the authors provide additional experiments and clarifications, after reading them, the reviewers and the meta-reviewer are still not fully convinced.
1. Reviewers are concerned about the privacy of the proposed method. The statement regarding privacy is not rigorous and only provides some vague intuitions without theoretical guarantees such as differential privacy.
1. The greedy selection algorithm is a heuristic and why it targets and can approximately solve a maximum coverage problem (the coverage is not well defined as in Eq. 3) is not clear. A rigorous optimization problem formulation and a straightforward connection to the greedy algorithm need to be presented
1. Some other major claims in the paper are not rigorous and might be misleading, e.g., the linear correlation between the degree of heterogeneity and model performance.
1. The original draft lacks experiments on different non-IID settings, >10-client setting, public dataset of comparable size as the local datasets, variance of the performance under different random seeds. Although the author provided partial results in the rebuttal, they did not cover the main scope of the experiments presented in the original draft.
1. The math notations in the paper are difficult to understand and need further corrections and simplifications.
1. FedDCA does not always show advantages, as reflected by the additional experiments during the discussion.
1. It is necessary to compare FedDCA with existing clustered federated learning approaches (with or without using the public dataset).

## Decision:
The authors provided detailed responses to the reviewers' concerns with additional experimental results and further clarifications. Authors and all the reviewers actively participate in the discussion. Although some concerns have been successfully addressed by the rebuttal, reviewers found that several major concerns above are still not resolved and may undermine the proposed setting and approach. The mete-reviewer carefully read all the discussion and responses, as well as the original paper. The meta-reviewer agrees with most reviewers that the paper is not ready for publication.

Based on the review comments, the author rebuttal, and the author-reviewer discussion, and the final review ratings, I cannot recommend acceptance of this paper to ICLR. The authors are encouraged to address the remaining concerns of the reviewers, clarify/justify the motivations of the problem settings better, formulate the problem and derive the algorithms in a more rigorous manner, and provide a fine-grained ablation study to demonstrate the advantages of the proposed methods and submit it to the next conference.

**Additional Comments On Reviewer Discussion:**

Authors provided detailed responses to the reviewers' concerns with additional experimental results and further clarifications. Authors and all the reviewers actively participate in the discussion. Although some concerns have been successfully addressed by the rebuttal, reviewers found that several major concerns above are still not resolved and may undermine the proposed setting and approach. The mete-reviewer carefully read all the discussion and responses, as well as the original paper. The meta-reviewer agrees with most reviewers that the paper is not ready for publication.

---

### Decision · Program_Chairs · 2025-01-22

Reject